# Gradients of Functions of Large Matrices

**Nicholas Krämer, Pablo Moreno-Muñoz, Hrittik Roy, Søren Hauberg**
Technical University of Denmark
Kongens Lyngby, Denmark
{pekra, pabmo, hroy, sohau}@dtu.dk

## Abstract

Tuning scientific and probabilistic machine learning models – for example, partial differential equations, Gaussian processes, or Bayesian neural networks – often relies on evaluating functions of matrices whose size grows with the data set or the number of parameters. While the state-of-the-art for *evaluating* these quantities is almost always based on Lanczos and Arnoldi iterations, the present work is the first to explain how to *differentiate* these workhorses of numerical linear algebra efficiently. To get there, we derive previously unknown adjoint systems for Lanczos and Arnoldi iterations, implement them in JAX, and show that the resulting code can compete with Diffrax when it comes to differentiating PDEs, GPyTorch for selecting Gaussian process models and beats standard factorisation methods for calibrating Bayesian neural networks. All this is achieved without any problem-specific code optimisation. Find the code at https://github.com/pnkraemer/experiments-lanczos-adjoints and install the library with pip install matfree.

## 1 Introduction

Automatic differentiation has dramatically altered the development of machine learning models by allowing us to forego laborious, application-dependent gradient derivations. The essence of this automation is to evaluate Jacobian-vector and vector-Jacobian products without ever instantiating the full Jacobian matrix, whose column count would match the number of parameters of the neural network. Nowadays, everyone can build algorithms around matrices of unprecedented sizes by exploiting this *matrix-free* implementation. However, differentiable linear algebra for Jacobian-vector products and similar operations has remained largely unexplored to this day. *We introduce a new matrix-free method for automatically differentiating functions of matrices.* Our algorithm yields the exact gradients of the forward pass, all gradients are obtained with the same code, and said code runs in linear time- and memory-complexity.

For a parametrised matrix $A = A(\theta) \in \mathbb{R}^{N \times N}$ and an analytic function $f : \mathbb{R} \to \mathbb{R}$, we call $f(A)$ a function of the matrix (different properties of $A$ imply different definitions of $f(A)$; one of them is applying $f$ to each eigenvalue of $A$ if $A$ is diagonalisable; see [1]). However, we assume that $A$ is the Jacobian of a large neural network or a matrix of similar size and never materialise $f(A)$. Instead, we only care about the values and gradients of the matrix-function-vector product

$$(\theta, v) \mapsto f[A(\theta)]v \tag{1}$$

assuming that $A$ is only accessed via differentiable matrix-vector products. Table 1 lists examples.

Evaluating Equation 1 is crucial for building large machine learning models, e.g., Bayesian neural networks: A common hyperparameter-calibration loss of a (Laplace-approximated) Bayesian neural network involves the log-determinant of the generalised Gauss–Newton matrix [13]

$$A(\alpha) \coloneqq \sum_{(x_i, y_i) \in \text{data}} [D_\theta g](x_i)^\top [D_g^2 \rho](y_i, g(x_i))[D_\theta g](x_i) + \alpha^2 I, \tag{2}$$

38th Conference on Neural Information Processing Systems (NeurIPS 2024).

Table 1: Some applications for functions of matrices. Log-determinants apply by combining $\log \det(A) = \text{trace}\,(\log(A))$ with stochastic trace estimation, which is why most vectors in this table are Rademacher samples. "PDE" / "ODE" = "Partial/Ordinary differential equation".

| Application | Function $f$ | Matrix $A$ | Vector $v$ | Parameter $\theta$ |
|---|---|---|---|---|
| PDEs & flows [2–5] | $e^\lambda$ | PDE discret. | PDE initial value | PDE |
| Gaussian process [6–8] | $\log(\lambda)$ | Kernel matrix | $v \sim$ Rademacher | Kernel |
| Invert. ResNets [9, 10] | $\log(1 + \lambda)$ | Jacobian matrix | $v \sim$ Rademacher | Network |
| Gaussian sampler [11] | $\sqrt{\lambda}$ | Covariance matrix | $v \sim N(0, I)$ | Covariance |
| Neural ODE [12] | $\lambda^2$ | Jacobian matrix | $v \sim$ Rademacher | Network |

where $D_\theta g$ is the parameter-Jacobian of the neural network $g$, $D_g^2 \rho$ is the Hessian of the loss function $\rho$ with respect to $g(x_i)$, and $\alpha$ is a to-be-tuned parameter. The matrix $A(\alpha)$ in Equation 2 has as many rows and columns as the network has parameters, which makes traditional, cubic-complexity linear algebra routines for log-determinant estimation entirely unfeasible. To compute this log-determinant, one chooses between either (i) simplifying the problem by pretending that the Hessian matrix is more structured than it actually is, e.g., diagonal [14]; or (ii) approximating $\log \det(A)$ by combining stochastic trace estimation [15]

$$\text{trace}\,(A) = \mathbb{E}\left[v^\top A v\right] \approx \frac{1}{L}\sum_{\ell=1}^{L} v_\ell^\top A v_\ell, \quad \text{for} \quad \mathbb{E}\left[vv^\top\right] = I, \tag{3}$$

with a Lanczos iteration $A(\theta) \approx QHQ^\top$ [16], to reduce the log-determinant to [17, 18]

$$\log \det(A) = \text{trace}\,(\log A) \approx \frac{1}{L}\sum_{\ell=1}^{L} v_\ell^\top \log(A) v_\ell \approx \frac{1}{L}\sum_{\ell=1}^{L} v_\ell^\top Q \log(H) Q^\top v_\ell. \tag{4}$$

The matrix $H$ in $A \approx QHQ^\top$ has as many rows/columns as we are willing to evaluate matrix-vector products with $A$; thus, it is small enough to evaluate the matrix-logarithm $\log(H)$ in cubic complexity.

**Contributions** This article explains how to differentiate not just log-determinants but any Lanczos and Arnoldi iteration so we can build loss functions for large models with such matrix-free algorithms (thereby completing the pipeline in Figure 1). This kind of functionality has been sorely missing from the toolbox of differentiable programming until now, even though the demand for functions of matrices is high in all of probabilistic and scientific machine learning [e.g. 2–12, 19–31].

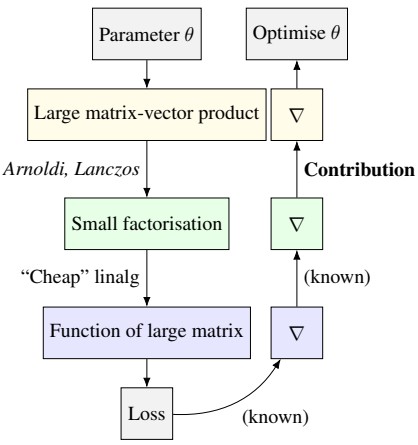

Figure 1: Values (down) and gradients (up) of functions of large matrices.

## 2 Related work

Here, we focus on applications in machine learning and illustrate how prior work avoids differentiating matrix-free decomposition methods like the Lanczos and Arnoldi iterations. Golub and Meurant [32] discuss applications outside machine learning.

*Generative models*, e.g., normalising flows [27, 33], rely on the change-of-variables formula, which involves the log-determinant of the Jacobian matrix of a neural network. Behrmann et al. [9] and Chen et al. [10] combine stochastic trace estimation with a Taylor-series expansion for the matrix logarithm. Ramesh and LeCun [34] use Chebyshev expansions instead of Taylor expansions. That said, Ubaru et al. [17] demonstrate how both methods converge more slowly than the Lanczos iteration when combined with stochastic trace estimation.

*Gaussian process model selection* requires values and gradients of log-probability density functions of Gaussian distributions (which involve log-determinants), where the covariance matrix $A(\theta)$ has as many rows and columns as there are data points [35]. Recent work [6, 7, 11, 19, 23, 36] all uses some

combination of stochastic trace estimation with the Lanczos iteration, and unanimously identifies gradients of log-determinants as ("d" shall be an infinitesimal perturbation; see Section 4)

$$\mu := \log \det(K(\theta)), \quad \mathrm{d}\mu = \operatorname{trace}\left(K(\theta)^{-1}\mathrm{d}K(\theta)\right). \tag{5}$$

Another round of stochastic trace estimation then estimates $\mathrm{d}\mu$ [6, 23, 36]. In contrast, our contribution is more fundamental: not only do we derive the exact gradients of the forward pass, but our formulation also applies to, say, matrix exponentials, whereas Equation 5 only works for log-determinants. Section 5 shows how our black-box gradients match state-of-the-art code for Equation 5 [6].

*Laplace approximations and neural tangent kernels* face the same problem of computing derivatives of log-determinants but with the generalised Gauss–Newton (GGN) matrix from Equation 2. In contrast to the Gaussian process literature, prior work on Laplace approximations prefers structured approximations of the GGN by considering subsets of network weights [37–39], or algebraic approximations of the GGN via diagonal, KFAC, or low-rank factors [31, 40–44]. All such approximations imply simple expressions for log-determinants, which are straightforward to differentiate automatically. Unfortunately, these approximations discard valuable information about the correlation between weights, so a linear-algebra-based approach leads to superior likelihood calibration (Section 7).

*Linear differential equations*, for instance $\dot{y}(t) = Ay(t)$, $y(0) = y_0$ are solved by matrix exponentials, $y(t) = \exp(At)y_0$. By this relation, matrix exponentials have frequent applications not just for the simulation of differential equations [e.g. 2, 45], but also for the construction of exponential integrators [3, 26, 29], state-space models [5, 46], and in generative modelling [4, 26, 28, 47]. There are many ways of computing matrix exponentials [48, 49], but only Al-Mohy and Higham [50] consider the problem of differentiating it and only in forward mode. In contrast, differential equations have a rich history of adjoint methods [e.g. 51, 52] with high-performance open-source libraries [53–56]. Still, the (now differentiable) Arnoldi iteration can compete with state-of-the-art solvers in JAX (Section 6).

## 3  Problem statement

Recall $A = A(\theta) \in \mathbb{R}^{N \times N}$ from Section 1. The focus of this paper is on matrices that are too large to store in memory, like Jacobians of neural networks or discretised partial differential equations:

**Assumption 3.1.** $A(\theta)$ *is only accessed via differentiable matrix-vector products* $(\theta, v) \mapsto A(\theta)v$.

The *de-facto* standard for linear algebra under Assumption 3.1 are matrix-free algorithms [e.g. 57, Chapters 10 & 11], like the conjugate gradient method for solving large sparse linear systems [58]. But there is more to matrix-free linear algebra than conjugate gradient solvers: Matrix-free implementations of matrix decompositions usually revolve around variations of the Arnoldi iteration [59], which takes an initial vector $v \in \mathbb{R}^N$ and a prescribed number of iterations $K \in \mathbb{N}$ and produces a column-orthogonal $Q \in \mathbb{R}^{N \times K}$, structured $H \in \mathbb{R}^{K \times K}$, residual vector $r \in \mathbb{R}^N$, and length $c \in \mathbb{R}$ such that

Figure 2: Lanczos/Arnoldi iteration.

$$AQ = QH + r(e_K)^\top, \quad \text{and} \quad Qe_1 = cv \tag{6}$$

hold (Figure 2; $e_1, e_K \in \mathbb{R}^K$ are the first and last unit vectors). If $A$ is symmetric, $H$ is tridiagonal, and the Arnoldi iteration becomes the *Lanczos iteration* [16]. Both iterations are popular for implementing matrix-function-vector products in a matrix-free fashion [1, 57], because the decomposition in Equation 6 implies $A \approx QHQ^\top$, thus

$$(\theta, v) \mapsto f(A(\theta))v \approx Qf(H)Q^\top v = c^{-1}Qf(H)e_1. \tag{7}$$

The last step, $Q^\top v = c^{-1}e_1$, is due to the orthogonality of $Q$. Since the number of matrix-vector products $K$ rarely exceeds a few hundreds or thousands, the following Assumption 3.2 is mild:

**Assumption 3.2.** *The map* $H \mapsto f(H)e_1$ *is differentiable, and* $Q$ *fits into memory.*

In summary, we evaluate functions of large matrices by firstly decomposing a large matrix into a product of small matrices (with Lanczos or Arnoldi) and, secondly, using conventional linear algebra to evaluate functions of small matrices. Functions of small matrices can already be differentiated efficiently [60–62]. *This work contributes gradients of the Lanczos and Arnoldi iteration under Assumptions 3.1 and 3.2, and thereby makes matrix-free implementations of matrix decompositions and functions of large matrices (reverse-mode) differentiable.*

Automatic differentiation, i.e. "backpropagating through" the matrix decomposition, is far too inefficient to be a viable option (Figure 3; setup in Appendix A). Our approach via implicit differentiation or the adjoint method, respectively, leads to gradients that inherit the linear runtime and memory-complexity of the forward pass.

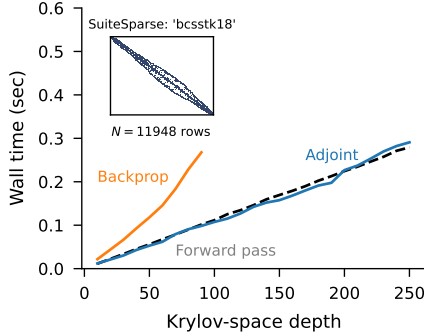

**Limitations and future work** The landscape of Lanczos- and Arnoldi-style matrix decompositions is vast, and some adjacent problems cannot be solved by this single article: (i) Forward-mode derivatives would require a derivation separate from what comes next. Yet, since functions of matrices map many to few parameters (matrices to vectors), reverse-mode is superior to forward-mode anyway [66, p. 153]. (ii) We only consider real-valued matrices

Figure 3: Backpropagation vs our adjoint method on a sparse matrix [63–65].

(for their relevance to machine learning), even though the decompositions generalise to complex arithmetic with applications in physics [67]. (iii) We assume $Q$ fits into memory, which relates to combining Arnoldi/Lanczos with full reorthogonalisation [57, 68–71]. Relaxing this assumption requires gradients of partial reorthogonalisation (among other things), which we leave to future work.

## 4 The method: Adjoints of the Lanczos and Arnoldi iterations

Numerical algorithms are rarely differentiated automatically, and usually, some form of what is known as "implicit differentiation" [66, 72] applies. The same is true for the Lanczos and Arnoldi iterations. However, and perhaps surprisingly, we differentiate the iterations like a dynamical system using the "adjoint method" [51, 73, 74], a variation of implicit differentiation that uses Lagrange multipliers [66], and not like a linear algebra routine [60, 61, 75]. To clarify this distinction, we briefly review implicit differentiation before the core contributions of this work in Sections 4.1 and 4.2.

**Notation** Let $\mathrm{d}x$ be an infinitesimal perturbation of some $x$. $D$ is the Jacobian operator, and $\langle \cdot, \cdot \rangle$ the Euclidean inner product between two equally-sized inputs. For a loss $\rho \in \mathbb{R}$ that depends on some $x$, the linearisation $\mathrm{d}\rho = D_x\rho\,\mathrm{d}x$ and the gradient identity $\mathrm{d}\rho = \langle \nabla_x\rho, \mathrm{d}x \rangle$ will be important [76, 77].

**Implicit differentiation** Let $\mathbf{a} : \theta \mapsto x$ be a numerical algorithm that computes some $x$ from some $\theta$. Assume that the input and output of $\mathbf{a}(\cdot)$ satisfy the constraint $\mathbf{c}(\theta, \mathbf{a}(\theta)) = 0$. For instance, if $\mathbf{a}(\cdot)$ solves $Ax = b$, the constraint is $\mathbf{c}(A, b; x) = Ax - b$ with $\theta := \{A, b\}$. We can use $\mathbf{c}(\cdot)$ in combination with the chain rule to find the derivatives of $\mathbf{a}(\cdot)$, ($\mathbf{c} = 0$ implies $\mathrm{d}\mathbf{c} = 0$)

$$0 = \mathrm{d}\mathbf{c}(\theta, x) = D_x\mathbf{c}(\theta, x)\mathrm{d}x + D_\theta\mathbf{c}(\theta, x)\mathrm{d}\theta. \tag{8}$$

In other words, we "linearise" the constraint $\mathbf{c}(\theta, x) = 0$. The adjoint method [78] proceeds by "transposing" this linearisation as follows. Let $\rho$ be a loss that depends on $y$ with gradient $\nabla_y\rho$ and recall the gradient identity from the "Notation" paragraph above. Then, for all Lagrange multipliers $\lambda$ with the same shape as the outputs of $\mathbf{c}(\cdot)$, we know that since $\mathbf{c} = 0$ implies $\mathrm{d}\mathbf{c} = 0$,

$$\mathrm{d}\rho = \langle \nabla_x\rho, \mathrm{d}x \rangle = \langle \nabla_x\rho, \mathrm{d}x \rangle + \langle \lambda, \mathrm{d}\mathbf{c} \rangle = \langle \nabla_x\rho + (D_x\mathbf{c})^\top\lambda, \mathrm{d}x \rangle + \langle (D_\theta\mathbf{c})^\top\lambda, \mathrm{d}\theta \rangle \tag{9}$$

must hold. By matching Equation 9 to $\mathrm{d}\rho = \langle \nabla_\theta\rho, \mathrm{d}\theta \rangle$ (this time, regarding $\rho$ as a function of $\theta$, not of $x$; recall the "Notation" paragraph), we conclude that if $\lambda$ solves the adjoint system

$$\nabla_x\rho + (D_x\mathbf{c})^\top\lambda = 0, \tag{10}$$

then $\nabla_\theta\rho := (D_\theta\mathbf{c})^\top\lambda$ must be the gradient of $\rho$ with respect to input $\theta$. This is the adjoint method [66, Section 10.4]. In automatic differentiation frameworks like JAX [79], this gradient implements a vector-Jacobian product with the Jacobian of $\mathbf{a}(\cdot)$ – implicitly via the Lagrange multiplier $\lambda$, without differentiating "through" $\mathbf{a}(\cdot)$ explicitly. In comparison to approaches that explicitly target vector-Jacobian products with implicit differentiation [like 66, Proposition 10.1], the adjoint method shines when applied to highly structured, non-vector-valued constraints, such as dynamical systems or the Lanczos and Arnoldi iterations. The reason is that the adjoint method does not change if $\mathbf{c}(\cdot)$ becomes matrix- or function-space-valued, as long as we can define inner products and adjoint operators, whereas other approaches (like what Blondel et al. [72] use for numerical optimisers) would become increasingly laborious in these cases. In summary, to reverse-mode differentiate a

numerical algorithm with the adjoint method, we need four steps: (i) find a constraint, (ii) linearise it, (iii) introduce Lagrange multipliers, and (iv) solve the resulting adjoint system. Carrying out those four steps for the Lanczos and Arnoldi iterations is the main contribution of the paper: Section 4.1 states both adjoint systems and Section 4.2 covers a matrix-free implementation.

## 4.1 Adjoint system of the Arnoldi and Lanczos iterations

Let $e_j$ be the $j$th unit vector. Denote by "$\circ$" the element-wise matrix product, and define the matrices

$$I_{\leq} := [\delta_{i \leq j}]_{i,j=1}^{K}, \quad I_{<} := [\delta_{i<j}]_{i,j=1}^{K}, \quad I_{\ll} := [\delta_{i+1<j}]_{i,j=1}^{K}, \tag{11}$$

so that for example, $I_{\leq} \circ A$ extracts the lower triangular matrix of $A$ (including the diagonal), and $I_{\ll} \circ A = 0$ enforces Hessenberg form [57]. The following two theorems do not require Assumptions 3.1 and 3.2, which are only relevant for analysing the computational complexities.

**Theorem 4.1** (Adjoint system of the Arnoldi iteration). *Let $K \in \mathbb{N}$, $v \in \mathbb{R}$, and $A \in \mathbb{R}^{N \times N}$, and a loss $\rho(\cdot) \in \mathbb{R}$ be given. If $Q \in \mathbb{R}^{N \times K}$, $H \in \mathbb{R}^{K \times K}$, $r \in \mathbb{R}^N$, and $c \in \mathbb{R}$ solve the* forward constraint

$$AQ = QH + r(e_K)^{\top}, \quad Qe_1 = vc, \quad I_{\leq} \circ [Q^{\top}Q] = I, \quad I_{\ll} \circ H = 0, \quad Q^{\top}r = 0, \tag{12}$$

*and if $\lambda \in \mathbb{R}^N$, $\Lambda \in \mathbb{R}^{N \times K}$, $\gamma \in \mathbb{R}^K$, $\Gamma \in \mathbb{R}^{K \times K}$, and $\Sigma \in \mathbb{R}^{K \times K}$ satisfy the* adjoint system

$$0 = \nabla_Q \rho + A^{\top}\Lambda - \Lambda H^{\top} + \lambda(e_1)^{\top} + Q(I_{\leq} \circ \Gamma) + Q(I_{\leq} \circ \Gamma)^{\top} + r\gamma^{\top} \tag{13a}$$

$$0 = \nabla_H \rho - Q^{\top}\Lambda + I_{\ll} \circ \Sigma \tag{13b}$$

$$0 = \nabla_r \rho - \Lambda e_K + Q\gamma \tag{13c}$$

$$0 = \nabla_c \rho - v^{\top}\lambda, \tag{13d}$$

*then the gradients of $\rho$ with respect to $A$ and $v$ are*

$$\nabla_A \rho := \Lambda Q^{\top}, \quad \nabla_v \rho := \lambda c. \tag{14}$$

*Sketch of the proof.* To derive the statement, start with Equation 12 as $\mathbf{c}(\cdot)$. Apply the chain- and product rules liberally to get $d\mathbf{c}(\cdot)$. Introduce Lagrange multipliers $\lambda$, $\Lambda$, $\gamma$, $\Gamma$, and $\Sigma$ like in the previous section, by adding Lagrange-multiplied constraints to

$$d\rho = \langle \nabla_Q \rho, dQ \rangle + \langle \nabla_H \rho, dH \rangle + \langle \nabla_r \rho, dr \rangle + \langle \nabla_c \rho, dc \rangle, \tag{15}$$

and rearrange the terms to see that Equation 13 implies Equation 14. Details are in Appendix B. $\square$

**Theorem 4.2** (Adjoint system of the Lanczos iteration). *Let a symmetric $A \in \mathbb{R}^{N \times N}$, as well as $v \in \mathbb{R}^N$, $K \in \mathbb{N}$, and a loss $\rho$ be known. In the following equations, set $b_0 := 1 \in \mathbb{R}$, $x_0 := 0 \in \mathbb{R}^n$, $\lambda_{K+1} := 0$, $\mu_0 := 0$, $\nu_0 := 0$ to simplify the expressions. If $x_1, ..., x_{K+1} \in \mathbb{R}^N$, and $a_1, ..., a_k, b_1, ..., b_k \in \mathbb{R}^K$, satisfy the* forward constraint

$$x_1 - v/(v^{\top}v) = 0, \tag{16a}$$

$$-b_{k-1}x_{k-1} + (A - a_k I)x_k - b_k x_{k+1} = 0, \qquad k = 1, ..., K \tag{16b}$$

$$x_{k+1}^{\top} x_{k+1} - 1 = 0, \qquad k = 1, ..., K, \tag{16c}$$

$$x_{k-1}^{\top} x_k = 0, \qquad k = 2, ..., K+1 \tag{16d}$$

*and if $\lambda_0, ..., \lambda_K \in \mathbb{R}^N$, $\mu_1, ..., \mu_K, \nu_1, ..., \nu_K \in \mathbb{R}$ satisfy the* adjoint system

$$0 = -\lambda_K b_K + (\nabla_{x_{K+1}} \rho + \mu_K x_{K+1} + \nu_K x_K), \tag{17a}$$

$$0 = -b_k \lambda_{k+1} + (A^{\top} - a_k I)\lambda_k - b_{k-1}\lambda_{k-1} + (\nabla_{x_k} \rho + \mu_{k-1} x_k + \nu_k x_{k+1} + \nu_{k-1} x_{k-1}), \tag{17b}$$

$$0 = \nabla_{a_k} \rho - \lambda_k^{\top} x_k, \tag{17c}$$

$$0 = \nabla_{b_k} \rho - \lambda_{k+1}^{\top} x_k - \lambda_k^{\top} x_{k+1}. \tag{17d}$$

*where all expressions involving $k$ hold for all $k = K, ..., 1$, then*

$$\nabla_v \rho := \frac{\lambda_0^{\top} x_1}{v^{\top}v} x_1 - \lambda_0, \quad \nabla_A \rho := \sum_{k=1}^{K} \lambda_k x_k^{\top} \tag{18}$$

*are the gradients of $\rho$ with respect to $v$ and $A$.*

*Sketch of the proof.* This theorem is proven similarly to that of Theorem 4.1, but instead of a few equations involving matrices, we have many equations involving scalars because for symmetric matrices, $H$ must be tridiagonal [57], and we expand $AQ = QH + r(e_{K+1})^\top$ column-wise. The coefficients $a_k$ and $b_k$ are the tridiagonal elements in $H$. We rename $q_k$ from Arnoldi to $x_k$ for Lanczos to make it easier to distinguish the two different sets of constraints. Details: Appendix C.  □

## 4.2 Matrix-free implementation

**Solving the adjoint systems** To compute $\nabla_A \rho$ and $\nabla_v \rho$, we need to solve the adjoint systems. When comparing the forward constraints to the adjoint systems, similarities emerge: for instance, the adjoint system of the Arnoldi iteration follows the same $A^{(\top)}H - \Lambda H^{(\top)} + \text{rest} = 0$ structure as the forward constraint. This structure suggests deriving a recursion for the backward pass that mirrors that of the forward pass. Appendix E contains this derivation and contrasts the resulting algorithm with that of the forward pass. The main observation is that the complexity of the adjoint passes for Lanczos and Arnoldi mirrors that of the forward passes. Gradients can be implemented purely with matrix-vector products, which is helpful because it makes our custom backward pass as matrix-free as backpropagation "through" the forward pass would be. This matrix-free implementation in combination with the efficient recursions in Theorems 4.1 and 4.2 explains the significant performance gains of our method compared to naive backpropagation, observed in Figure 3.

Theorems 4.1 and 4.2's expressions for $\nabla_A \rho$ are not directly applicable when we only have matrix-vector products with $A$. Fortunately, parameter-gradients emerge from matrix-gradients:

**Corollary 4.3** (Parameter gradients). *Under Assumption 3.1 and the assumptions of Theorem 4.1, and if $A$ is parametrised by some $\theta$, the gradients of $\rho$ with respect to $\theta$ are*

$$\nabla_\theta \rho = \sum_{k=1}^{K} \nabla \left[ \theta \mapsto (e_k)^\top Q^\top A(\theta)^\top \Lambda e_k \right], \tag{19}$$

*which can be assembled online during the backward pass. For the Lanczos iteration, we assume the conditions of Theorem 4.2 instead of Theorem 4.1, replace $Qe_k$ and $\Lambda e_k$ with $x_k$ and $\lambda_k$, let the sum run from $k = 0$ to $k = K$, and the rest of this statement remains true.*

*Sketch of the proof.* The proof of this identity combines the expression(s) for $\nabla_A \rho$ from Theorems 4.1 and 4.2 with $\mathrm{d}A = D_\theta A \mathrm{d}\theta$. The derivations are lengthy and therefore relegated to Appendix D.  □

**Reorthogonalisation** It is well known that the Lanczos and Arnoldi iterations suffer from a loss of orthogonality and that reorthogonalisation of the columns in $Q$ is often necessary [68–71]. Reorthogonalisation does not affect the forward constraints, so the adjoint systems remain the same with and without reorthogonalisation. But adjoint systems also suffer from a loss of orthogonality: The equivalent of orthogonality for the adjoint system is the projection constraint in Equation 13b, which constrains the Lagrange multipliers $\Lambda$ to a hyperplane defined by $Q$ and other known quantities. The constraint can – and should (Table 2) – be used whenever the forward pass requires reorthogonalisation.[1] In the case studies below, we always use full reorthogonalisation on the forward and adjoint pass, also for the Arnoldi iteration [71, Table 7.1], even though this is slightly less common than for the Lanczos iteration.

Table 2: Accuracy loss when differentiating the Arnoldi iteration on a Hilbert matrix in double precision ($\phi$ : decompose with a full-rank Arnoldi iteration, then reconstruct the original matrix; measure $\|\partial \phi - I\|$; details in Appendix F).

|  | Loss of accuracy |
| --- | --- |
| Adjoint w/o proj. | $5.83 \cdot 10^{-3}$ |
| Adjoint w/ proj. | $\mathbf{1.17 \cdot 10^{-10}}$ |
| Backprop. | $\mathbf{1.17 \cdot 10^{-10}}$ |

**Summary (before the case studies)** The main takeaway from Sections 4.1 and 4.2 is that now, we do not only have closed-form expressions for the gradients of Arnoldi and Lanczos iterations (Theorems 4.1 and 4.2), but that we can compute them in the same complexity as the forward pass, in a numerically stable way, and evaluate parameter-gradients in linear time- and space-complexity (Corollary 4.3). While some of the derivations are somewhat technical, the overall

---

[1]The adjoint system of the Lanczos iteration does not admit this projection constraint, but we can implement re-orthogonalised Lanczos via calling the Arnoldi code. This induces only minimal overhead because fully reorthogonalised Lanczos code has roughly the same complexity as Arnoldi code.

Table 3: Our method yields the same root-mean-square errors (RMSEs) as GPyTorch. It reaches lower training losses but is $\approx 20\times$ slower per epoch due to different matrix-vector-product backends (see Appendix G). Three runs, significant improvements in bold. We use an 80/20 train/test split.

| Dataset | Size | Dim. | Method | RMSE ↓ | Final training loss ↓ | Runtime (s/epoch) ↓ |
|---|---|---|---|---|---|---|
| elevators | 16,599 | 18 | Adjoints | $0.09 \pm 0.002$ | **-0.91 ± 0.025** | $1.69 \pm 0.000$ |
| | | | GPyTorch | $0.09 \pm 0.003$ | $-0.63 \pm 0.062$ | **0.10 ± 0.004** |
| protein | 45,730 | 9 | Adjoints | $0.39 \pm 0.005$ | $0.73 \pm 0.300$ | $12.62 \pm 0.172$ |
| | | | GPyTorch | $0.39 \pm 0.005$ | $0.73 \pm 0.075$ | **0.73 ± 0.039** |
| kin40k | 40,000 | 8 | Adjoints | $0.12 \pm 0.004$ | **-0.30 ± 0.078** | $8.27 \pm 0.004$ |
| | | | GPyTorch | $0.10 \pm 0.010$ | $-0.26 \pm 0.094$ | **0.26 ± 0.024** |
| kegg_dir | 48,827 | 20 | Adjoints | $0.12 \pm 0.002$ | $-0.59 \pm 0.295$ | $13.25 \pm 0.005$ |
| | | | GPyTorch | $0.12 \pm 0.005$ | $-0.41 \pm 0.054$ | **0.62 ± 0.262** |
| kegg_undir | 63,608 | 26 | Adjoints | $0.12 \pm 0.002$ | **-0.69 ± 0.263** | $24.20 \pm 0.004$ |
| | | | GPyTorch | $0.12 \pm 0.003$ | $-0.40 \pm 0.039$ | **1.67 ± 0.532** |

approach follows the general template for the adjoint method relatively closely. The resulting algorithm beats backpropagation "through" the iterations by a margin in terms of speed (Figure 3) and enjoys the same stability gains from reorthogonalisation as the forward pass (Table 2). Our open-source implementation of reverse-mode differentiable Lanczos and Arnoldi iterations can be installed via "pip install matfree". Next, we put this code to the test on three challenging machine-learning problems centred around functions of matrices to see how it fares against state-of-the-art differentiable implementations of exact Gaussian processes (Section 5), differential equation solvers (Section 6), and Bayesian neural networks (Section 7).

## 5 Case study: Exact Gaussian processes

Model selection for Gaussian processes has arguably been the strongest proponent of the Lanczos iteration and similar matrix-free algorithms in recent years [6, 7, 11, 19, 20, 23, 80], and most of these efforts have been bundled up in the GPyTorch library [6]. For example, GPyTorch defaults to choosing a Lanczos iteration over a Cholesky decomposition as soon as the dataset exceeds 800 data points.[2] Calibrating hyperparameters of Gaussian process models involves optimising log-marginal-likelihoods of the regression targets, which requires computing $x^\top A^{-1} x$ and $\log \det(A)$ for a covariance matrix $A$ with as many rows and columns as there are data points. Recent works [6, 7, 11, 19, 20, 23, 80] unanimously suggest to differentiate log-determinants via $\mu := \text{trace}(\log(A))$ and $\mathrm{d}\mu = \text{trace}(A^{-1}\mathrm{d}A)$ (Equation 5). Since we seem to be the first to take a different path, benchmarking Gaussian processes in comparison to GPyTorch is a good first testbed for our gradients.

**Setup: Like GPyTorch's defaults** We mimic recent suggestions for scalable Gaussian process models [7, 19]: we implement a pivoted Cholesky preconditioner [81] and combine it with conjugate gradient solvers for $x^\top K^{-1} x$ (which can be differentiated efficiently). We estimate the log-determinant stochastically via $\log \det(A) = \text{trace}(\log(A)) = E[v^\top \log(A)v]$, and compute $\log(A)v$ via the Lanczos iteration. While all of the above is common for "exact" Gaussian processes [6, 7, 19] ("exact" as opposed to variational approaches, which are not relevant for this comparison), there are three key differences between our code and GPyTorch's: (i) GPyTorch is in Pytorch and uses KeOps [82] for efficient kernel-matrix-vector products. We use JAX and must build our own low-memory matrix-vector products (Appendix G). (ii) GPyTorch runs all algorithms adaptively (we specify tolerances and maximum iterations as much as possible). We use adaptive conjugate gradient solvers and fixed ranks for everything else. *(iii) GPytorch differentiates the log-determinant with a tailored approximation of Equation 5 [6]; we embed our gradients of the Lanczos iteration into automatic differentiation.* To keep the benchmark objective, we mimic the parameter suggestions from GPyTorch's default settings, and optimise hyperparameters of a Matérn$\left(\frac{3}{2}\right)$ model on UCI datasets with the Adam optimiser [83]. Appendix H lists parameters and discusses the datasets.

---

[2]Parameter `max_cholesky_size`: https://docs.gpytorch.ai/en/stable/settings.html.

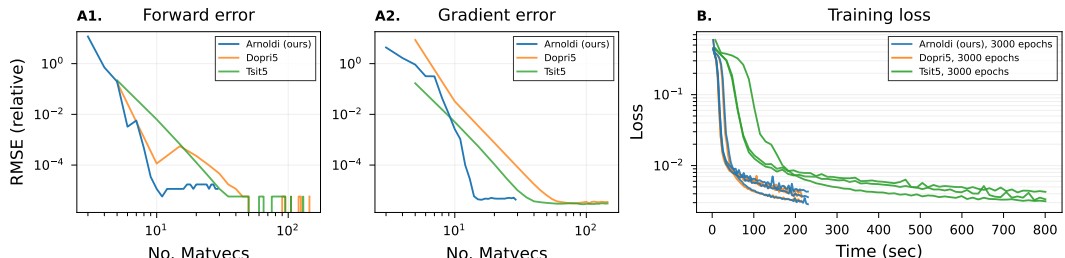

Figure 5: Arnoldi's superior convergence on the forward pass (A1) is inherited by the gradients (A2; mind the shared $y$-axis) and ultimately leads to fast training (B). For training, Arnoldi uses ten matrix-vector products, and the other two use 15 (so they have equal error $\approx 10^{-4}$ in A1 and A2.)

**Analysis: Trains like GPyTorch; large scale only limited by matrix-vector-product backends** In this benchmark, we are looking for low reconstruction errors, fast runtimes and well-behaved loss functions. Table 3 shows that this is the case for both implementations: the reconstruction errors are essentially the same, and both methods converge well (we achieve lower training losses). This result shows that by taking a numerically exact gradient of the Lanczos iteration, and leaving everything else to automatic differentiation, matches the performance of state-of-the-art solvers. Larger datasets are only limited by the efficiency of our matrix-vector products (in comparison to KeOps); Appendix G discusses this in detail. Overall, this result strengthens the democratisation of exact Gaussian processes because it reveils a simple yet effective alternative to GPyTorch's domain-specific gradients.

## 6 Case study: Physics-informed machine learning with PDEs

Much of the paper thus far discusses functions of matrices in the context of log-determinants. So, in order to demonstrate performance for (i) a problem that is not a log-determinant and (ii) for a non-symmetric matrix which requires Arnoldi instead of Lanczos, we learn the coefficient field $\omega$ of

$$\frac{\partial^2}{\partial t^2} u(t; x_1, x_2) = \omega(x_1, x_2)^2 \left[ \frac{\partial^2}{\partial x_1^2} u(t; x_1, x_2) + \frac{\partial^2}{\partial x_2^2} u(t; x_1, x_2) \right] \tag{20}$$

subject to Neumann boundary conditions. We discretise this equation on a $128 \times 128$ grid in space and transform the resulting $128^2$-dimensional second-order ordinary differential equation into a first-order differential equation, $\dot{w} = Aw$, $w(0) = w_0$, with solution operator $w(t) = \exp(At)w_0$. The system matrix $A$ is sparse, asymmetric, and has $32{,}768$ rows and columns. We sample a true $\omega$ from a Gaussian process with a square exponential kernel and generate data by sampling 256 initial conditions and solving the equation numerically with high precision. Details are in Appendix I.

**Setup: Arnoldi vs Diffrax's Runge-Kutta methods for a 250k parameter MLP** We learn $\omega$ with a multi-layer perceptron (MLP) with approximately 250,000 parameters. We had similar reconstructions with fewer parameters but use 250,000 to display how gradients of the Arnoldi iteration scale to many parameters. We compare an implementation of the solution operator $(\theta, w_0) \mapsto \exp(A(\theta))w_0$ with the Arnoldi iteration to Diffrax's [54] implementation of "Dopri5" [84, 85] with a differentiate-then-discretise adjoint [86] as well as "Tsit5" [87] with a discretise-then-differentiate adjoint (recommended by [53, 54]). All methods receive equal matrix-vector products per simulation.

**Analysis: All methods train, but Arnoldi is more accurate for fixed matrix-vector-product budgets** We evaluate the approximation errors in computing the values and gradients of a mean-squared error loss for all three solvers and then use the solvers to train the MLP. We are looking for low approximation errors for few matrix-vector products and for a good reconstruction of the truth. Figure 5 shows the results. The Arnoldi iteration has the lowest forward-pass and gradient error, but Table 4 demonstrates how all ap-

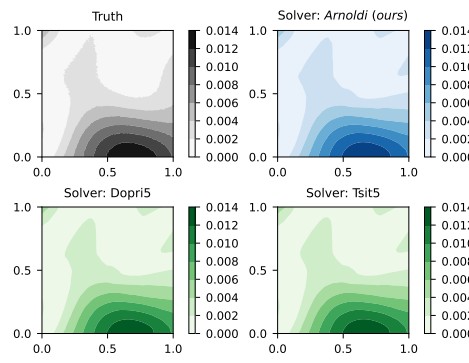

Figure 4: All methods find the truth.

Table 4: All three methods reconstruct the parameter well (std.-deviations exceed differences for test-loss and RMSE), but Arnoldi and Dopri5 are faster than Tsit5. Dopri5 uses the `BacksolveAdjoint`, and Tsit5 the `RecursiveCheckpointAdjoint` in Diffrax [54]. We contribute Arnoldi's adjoints.

|  | Arnoldi (adjoints; ours) | Dopri5 (diff. → disc.) | Tsit5 (disc. → diff.) |
|---|---|---|---|
| Loss on test set | 6.1e-03 $\pm$ 3.3e-04 | 6.3e-03 $\pm$ 5.7e-04 | 5.9e-03 $\pm$ 2.2e-04 |
| Parameter RMSE | 2.9e-04 $\pm$ 4.4e-05 | 2.6e-04 $\pm$ 5.0e-05 | 2.7e-04 $\pm$ 5.2e-05 |
| Runtime per epoch | **7.7e-02 $\pm$ 1.8e-05** | **7.2e-02 $\pm$ 3.4e-05** | 2.7e-01 $\pm$ 1.1e-05 |

proaches lead to low errors on $\omega$ as well as on a test set (a held-back percentage of the training data); see also Figure 4. The adjoints of the Arnoldi iteration match the efficiency of the differentiate-then-discretise adjoint [86], and both outperform the discretise-then-differentiate adjoint by a margin. This shows how linear-algebra solutions to matrix exponentials can compete with highly optimised differential equation solvers. We anticipate ample opportunities of using the now-differentiable Arnoldi iteration for physics-based machine learning.

## 7 Case study: Calibrating Bayesian neural networks

Next, we differentiate a function of a matrix on a problem that is native to machine learning: marginal likelihood optimisation of a Bayesian neural network (a high-level introduction is in Appendix J).

**Setup: Laplace-approximation of a VAN pre-trained on ImageNet** We consider as $g_\theta(x)$ a "Visual Attention Network" [88] with 4,105,800 parameters, pre-trained on ImageNet [89]. We assume $p(\theta) = N(0, \alpha^{-2}I)$, and Laplace-approximate the log-marginal likelihood of the data as

$$\log p(y \mid x) \approx \log p(y, \theta \mid x) - \frac{1}{2} \log \det(A(\alpha)) + \text{const} \tag{21}$$

where $A(\alpha)$ is the generalised Gauss–Newton matrix (GGN) from Section 1 (recall Equation 2). We optimise $\alpha$ via Equation 21, implementing the log-determinant via stochastic trace estimation in combination with a Lanczos iteration (like in Section 5). Contemporary works [31, 37–43] rely on sparse approximations of the GGN (such as diagonal or KFAC approximations), so we compare our implementation to a diagonal approximation of the GGN matrix, which yields closed-form log-determinants. The exact diagonal of the 4-million-column GGN matrix would require 4 million GGN-vector products with unit vectors, and like Deng et al. [90], we find this too expensive and resort to stochastic diagonal approximation (similar to trace estimation; all details are in Appendix J). We give both the stochastic diagonal approximation and our Lanczos-based estimator exactly 150 matrix-vector products to approximate Equation 21. We compare the evolution of the loss function over time and various uncertainty calibration metrics. Figure 6 demonstrates training and Table 5 shows results.

Figure 6: Lanczos vs diagonal approx. for a Bayesian VAN.

**Analysis: Lanczos uses matrix-vector products better (by a margin)** The results suggest how, for a fixed matrix-vector-product budget, Lanczos achieves a drastically better likelihood at a similar computational budget and already shows significant improvement with a much smaller budget. Lanczos outperforms the diagonal approximation on all metrics except ECE. The subpar performance of the diagonal approximation matches the observations of Ritter et al. [31]; see also [14]. The main takeaway from this study is that differentiable matrix-free linear algebra unlocks new techniques for Laplace approximations and allows further advances for Bayesian neural networks in general.

Table 5: Lanczos outperforms the diagonal approximation for calibrating a Bayesian version of an ImageNet pre-trained VAN. One training run; calibration estimated with 30 samples (sampling "Lanczos" and "diagonal" with another Lanczos/diagonal approximation; see Appendix J). `places365` [91] is the out-of-distribution data; mean and std.-deviations of 3 runs. "MVs": matrix-vector products.

| | | Lanczos (50 MVs) | Lanczos (150 MVs) | Diagonal (150 MVs) |
|---|---|---|---|---|
| Runtime (sec) | ↓ | **950** | 4674 | 4314 |
| Marginal likelihood (log) | ↑ | -192,757 | **-154,475** | -876,444 |
| Joint likelihood: train (log) | ↑ | **-81.2 ± 15.4** | **-81.2 ± 15.4** | -5,669.2 ± 124.1 |
| Joint likelihood: test (log) | ↑ | **-66.5 ± 11.2** | **-66.5 ± 11.2** | -5,260.9 ± 290.2 |
| Expected calibration error | ↓ | 0.5 ± 0.01 | 0.5 ± 0.01 | **0.2 ± 0.003** |
| AUROC (out-of-dist.) | ↑ | **0.9 ± 0.03** | **0.9 ± 0.03** | 0.5 ± 0.010 |

## Acknowledgments and Disclosure of Funding

This work was supported by a research grant (42062) from VILLUM FONDEN. The work was partly funded by the Novo Nordisk Foundation through the Center for Basic Machine Learning Research in Life Science (NNF20OC0062606). This project received funding from the European Research Council (ERC) under the European Union's Horizon programme (grant agreement 101125993).

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

## Appendix: Overview

Some of the results in the main paper promised detailed information about setups, data, compute, or additional proofs. For example, the case study about partial differential equations involves a data generation process which will receive further explanation in this supplement.

The appendix provides the following details: Appendices A and F elaborate on Figure 3 and Table 2 respectively; Appendices B to E contain proofs for the main results; and Appendices G to J describe the setup used for the case studies. Notably, Appendix G describes how we implement low-memory matrix-vector products in JAX (to replicate what makes libraries like KeOps [82] so efficient), and Appendix I outlines a PDE data set similar to that by Liu et al. [92].

**Code** Most of the contributions of this paper pertain to differentiable implementations of numerical algorithms – at the heart of it are our reverse-mode differentiable Lanczos and Arnoldi iterations. We provide JAX code to reproduce all experiments at the URL

```
https://github.com/pnkraemer/experiments-lanczos-adjoints
```

and have packaged all numerical methods in a JAX library that can be installed via

```
pip install matfree
```

Next to Lanczos and Arnoldi, this includes variants of conjugate gradient methods [58] and pivoted Cholesky preconditioners [81] (which we used for Gaussian processes), low-memory kernel-matrix-vector products (also Gaussian processes), efficient GGN-vector products (used for Bayesian neural networks), and matrix-free sampling algorithms to sample from Gaussian processes and Laplace-approximated Bayesian neural networks (used for Gaussian processes and Bayesian neural networks). These methods are known in some form or another, but until now, they have all lacked a software implementation in the current JAX ecosystem (with some exceptions relating to conjugate gradients).

**Compute** All experiments before the case studies were run on CPU. The Gaussian process and differential equation case studies run on a V100 GPU, the Bayesian neural network one on a P100 GPU. The Gaussian process and Bayesian neural network studies run in a few hours, all other code finishes in a few minutes.

## A  Additional context for Figure 3

To create Figure 3, we load the "bcsstk18" matrix from the SuiteSparse matrix collection [63–65]. This matrix is symmetric and has 11,948 rows/columns and 149,090 nonzero entries. We implement matrix-vector products with `jax.experimental.sparse`, and use a Lanczos iteration without reorthogonalisation. We time the execution of the forward pass, as well as the backward pass with and without implementing a custom vector-Jacobian product (the custom vector-Jacobian product involves the adjoint). The results were shown in Figure 3, and displayed how rapidly the computational complexity of automatic differentiation increases, whereas the computational complexity of our custom gradient mirrors that of the forward pass. This figure showed how without our proposed gradients, differentiating through the Lanczos iteration is unfeasible.

To add to this benchmark, we repeat the same for the Arnoldi iteration and show both compilation and runtime for Lanczos and Arnoldi in Figure 7 (this includes the curves from Figure 3 again). We see

Figure 7: Run and compilation times for Lanczos and Arnoldi. The "backprop" curves were stopped at 100, because for higher values we encountered memory issues. All experiments run on CPU.

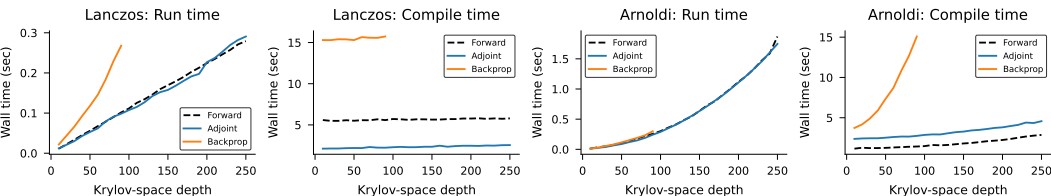

different behaviour for Lanczos and Arnoldi. Whereas for back-propagation through Lanczos without

the custom gradient, the compilation times remain constant for increasing Krylov-space depth $K$ and runtimes increase rapidly, the reverse is true for the Arnoldi iteration. The adjoint method mirrors that of the forward pass in both benchmarks. In either case, the increasing memory requirements for backpropagation through Lanczos and Arnoldi without our proposed adjoints becomes apparent.

## B    Proof of Theorem 4.1

Here is how we derive the adjoints of the Arnoldi system. The structure is the usual: take the forward constraint, differentiate, add Lagrange multipliers ("transpose"), and identify the adjoint system.

### B.1    Linearisation

The linearisation of Equation 12 is

$$(\mathrm{d}A)Q + A\mathrm{d}Q - (\mathrm{d}Q)H - Q\mathrm{d}H - (\mathrm{d}r)(e_K)^\top = 0 \quad \in \mathbb{R}^{N\times K}, \tag{22a}$$

$$(\mathrm{d}Q)e_1 - (\mathrm{d}v)c - v\mathrm{d}c = 0 \quad \in \mathbb{R}^{N\times 1}, \tag{22b}$$

$$I_\le \circ [\langle Qe_i, (\mathrm{d}Q)e_j \rangle + \langle Qe_j, (\mathrm{d}Q)e_i \rangle]_{i,j=1}^K = 0 \quad \in \mathbb{R}^{K\times K}, \tag{22c}$$

$$I_\ll \circ \mathrm{d}H = 0 \quad \in \mathbb{R}^{K\times K}, \tag{22d}$$

$$[\langle r, (\mathrm{d}Q)e_j \rangle + \langle Qe_j, \mathrm{d}r \rangle]_{j=1}^K = 0 \quad \in \mathbb{R}^{K\times 1}. \tag{22e}$$

To see this, apply the chain- and product rules to the original constraint in Equation 12.

### B.2    Transposition

Let $\rho = \rho(Q, r, H, c) \in \mathbb{R}$ be a scalar function of the outputs. In the following, interpret vectors as $N \times 1$ matrices and scalars as $1 \times 1$ matrices. The values of inner products and the realisations of $\rho$ are the only scalars.

For all $\Lambda \in \mathbb{R}^{N\times K}$, $\lambda \in \mathbb{R}^{N\times 1}$, $\Gamma, \Sigma \in \mathbb{R}^{K\times K}$, and $\gamma \in \mathbb{R}^{K\times 1}$, we have

$$\mathrm{d}\rho = \langle \nabla_Q\rho, \mathrm{d}Q \rangle + \langle \nabla_H\rho, \mathrm{d}H \rangle + \langle \nabla_r\rho, \mathrm{d}r \rangle + \langle \nabla_c\rho, \mathrm{d}c \rangle \tag{23a}$$

$$= \langle \nabla_Q\rho, \mathrm{d}Q \rangle + \langle \nabla_H\rho, \mathrm{d}H \rangle + \langle \nabla_r\rho, \mathrm{d}r \rangle + \langle \nabla_c\rho, \mathrm{d}c \rangle$$
$$+ \langle \Lambda, (\mathrm{d}A)Q + A\mathrm{d}Q - (\mathrm{d}Q)H - Q\mathrm{d}H - (\mathrm{d}r)(e_K)^\top \rangle$$
$$+ \langle \lambda, (\mathrm{d}Q)e_1 - (\mathrm{d}v)c - v\mathrm{d}c \rangle$$
$$+ \langle \Gamma, I_\le \circ [\langle Qe_i, (\mathrm{d}Q)e_j \rangle + \langle Qe_j, (\mathrm{d}Q)e_i \rangle]_{i,j=1}^K \rangle$$
$$+ \langle \Sigma, I_\ll \circ \mathrm{d}H \rangle$$
$$+ \langle \gamma, [\langle r, (\mathrm{d}Q)e_j \rangle + \langle Qe_j, \mathrm{d}r \rangle]_{j=1}^K \rangle \tag{23b}$$

$$= \langle \nabla_Q\rho, \mathrm{d}Q \rangle + \langle \nabla_H\rho, \mathrm{d}H \rangle + \langle \nabla_r\rho, \mathrm{d}r \rangle + \langle \nabla_c\rho, \mathrm{d}c \rangle$$
$$+ \langle \Lambda Q^\top, \mathrm{d}A \rangle + \langle A^\top\Lambda, \mathrm{d}Q \rangle - \langle \Lambda H^\top, \mathrm{d}Q \rangle - \langle Q^\top\Lambda, \mathrm{d}H \rangle - \langle \Lambda e_K, \mathrm{d}r \rangle$$
$$+ \langle \lambda(e_1)^\top, \mathrm{d}Q \rangle - \langle \lambda c^\top, \mathrm{d}v \rangle - \langle v^\top\lambda, \mathrm{d}c \rangle$$
$$+ \langle Q(I_\le \circ \Gamma), \mathrm{d}Q \rangle + \langle Q(I_\le \circ \Gamma)^\top, \mathrm{d}Q \rangle$$
$$+ \langle I_\ll \circ \Sigma, \mathrm{d}H \rangle$$
$$+ \langle r\gamma^\top, \mathrm{d}Q \rangle + \langle Q\gamma, \mathrm{d}r \rangle \tag{23c}$$

$$=: \langle Z_Q, \mathrm{d}Q \rangle + \langle Z_H, \mathrm{d}H \rangle + \langle Z_r, \mathrm{d}r \rangle + \langle Z_c, \mathrm{d}c \rangle + \langle \Lambda Q^\top, \mathrm{d}A \rangle + \langle \lambda c^\top, \mathrm{d}v \rangle \tag{23d}$$

with the constraints

$$Z_Q := \nabla_Q\rho + A^\top\Lambda - \Lambda H^\top + \lambda(e_1)^\top + Q(I_\le \circ \Gamma) + Q(I_\le \circ \Gamma)^\top + r\gamma^\top \quad \in \mathbb{R}^{N\times K} \tag{24a}$$

$$Z_H := \nabla_H\rho - Q^\top\Lambda + I_\ll \circ \Sigma \quad \in \mathbb{R}^{K\times K} \tag{24b}$$

$$Z_r := \nabla_r\rho - \Lambda e_K + Q\gamma \quad \in \mathbb{R}^{N\times 1} \tag{24c}$$

$$Z_c := \nabla_c\rho - v^\top\lambda \quad \in \mathbb{R}^{1\times 1}. \tag{24d}$$

Solving the adjoint system, $Z_Q = 0$, $Z_H = 0$, $Z_r = 0$, and $Z_c = 0$ as a function of $\Lambda, \lambda, \Gamma, \Sigma$, and $\gamma$, yields the desired $\nabla_A\rho = \Lambda Q^\top$ and $\nabla_v f = \lambda c^\top$. Theorem 4.1 is complete.

## C    Proof of Theorem 4.2

Recall that $\rho = \rho(x_1, ..., x_{K+1}; a_1, ..., a_K; b_1, ..., b_K)$ shall be a scalar/loss that depends on the output of the algorithm. Denote by $\nabla_{x_k}\rho$ the gradient of $\rho$ with respect to each Lanczos vector $x_k$, and by $\nabla_{a_k}\rho$ and $\nabla_{b_k}\rho$ the gradients with respect to $a_k$ and $b_k$ respectively.

The differential of normalisation, i.e., the operation $s \mapsto h = s/(s^\top s)$ is

$$\mathrm{d}h = \frac{1}{s^\top s}\left(I - hh^\top\right)\mathrm{d}s. \tag{25}$$

The next steps are the usual ones: we start with the forward constraint, linearise, add Lagrange multipliers, and identify the adjoint system. The order of the middle two steps (linearise, multipliers) is interchangeable; while for Arnoldi, we linearise first and then add Lagrange multipliers, for Lanczos, we go the other way.

Define Lagrange multipliers $\{\lambda_k\}_{k=0}^K \subseteq \mathbb{R}^n$ and $\{\mu_k/2, \nu_k\}_{j,k=1}^K \subseteq \mathbb{R}$,

$$\rho = \rho + \sum_{k=1}^K \lambda_k^\top \left(-b_{k-1}x_{k-1} + (A - a_k I)x_k - b_k x_{k+1}\right) \qquad (b_0 = 1, x_0 = 0)$$

$$- \lambda_0^\top\left(x_1 - \frac{v}{\sqrt{v^\top v}}\right) + \frac{1}{2}\sum_{k=1}^K \mu_k(x_{k+1}^\top x_{k+1} - 1) + \sum_{k=1}^K \nu_k\, x_k^\top x_{k+1}. \tag{26}$$

Differentiate and use that the forward constraint must be satisfied,

$$\mathrm{d}\rho = \sum_{k=1}^{K+1}(\nabla_{x_k}\rho)^\top \mathrm{d}x_k + \sum_{k=1}^K (\nabla_{a_k}\rho)^\top \mathrm{d}a_k + \sum_{k=1}^K (\nabla_{b_k}\rho)^\top \mathrm{d}b_k$$

$$+ \sum_{k=1}^K \lambda_k^\top \left[(\mathrm{d}A)x_k - (\mathrm{d}a_k)x_k - (\mathrm{d}b_{k-1})x_{k-1} - (\mathrm{d}b_k)x_{k+1}\right] \qquad (\mathrm{d}b_0 = 1, \mathrm{d}x_0 = 0)$$

$$+ \sum_{k=1}^K \lambda_k^\top \left[A\mathrm{d}x_k - a_k\mathrm{d}x_k - b_{k-1}\mathrm{d}x_{k-1} - b_k\mathrm{d}x_{k+1}\right] \qquad (b_0 = 1, x_0 = 0)$$

$$- \lambda_0^\top(\mathrm{d}x_1 - (I - x_1 x_1^\top)/(v^\top v)\mathrm{d}v)$$

$$+ \sum_{k=1}^K \mu_k\, x_{k+1}^\top \mathrm{d}x_{k+1}$$

$$+ \sum_{k=1}^K \nu_k(x_{k+1}^\top \mathrm{d}x_k + x_k^\top \mathrm{d}x_{k+1}). \tag{27}$$

Sort all terms by differential,

$$\mathrm{d}\rho = \sum_{k=1}^K Z_{a_k}\mathrm{d}a_k + \sum_{k=1}^K Z_{b_k}\mathrm{d}b_k + \sum_{k=1}^{K+1} Z_{x_k}\mathrm{d}x_k + Z_v\mathrm{d}v + \mathrm{trace}\left(Z_A\mathrm{d}A\right). \tag{28}$$

so that enforcing that all $Z_{a_k}, Z_{b_k}, Z_{x_k}$ terms are zero yields constraints for the multipliers from which we can compute gradients with respect to $v$ and $A$.

What are those terms? Let $\lambda_{K+1} = 0$, $\mu_0 = 0$, and $\nu_0 = 0$ (to simplify notation below); then,

$$Z_{x_{K+1}} = -\lambda_K b_K + (\nabla_{x_{K+1}}\rho + \mu_K x_{K+1} + \nu_K x_K), \tag{29}$$

and for all $k = K, ...1$,     (recall $x_0 = 0, b_0 = 1, \mu_0 = 0, \nu_0 = 0, \lambda_{K+1} = 0$)

$$Z_{x_k} = -b_k\lambda_{k+1} + (A^\top - a_k I)\lambda_k - b_{k-1}\lambda_{k-1} + (\nabla_{x_k}\rho + \mu_{k-1}x_k + \nu_k x_{k+1} + \nu_{k-1}x_{k-1}), \tag{30a}$$

$$Z_{a_k} = \nabla_{a_k}\rho - \lambda_k^\top x_k, \tag{30b}$$

$$Z_{b_k} = \nabla_{b_k}\rho - \lambda_{k+1}^\top x_k - \lambda_k^\top x_{k+1}. \tag{30c}$$

The expressions are like the forward-Lanczos constraints, and the main differences are (i) that the recursions are run backwards in "time" and (ii) the existence of a nonzero bias term in the adjoints (marked by parentheses). Enforcing $Z_{a_k}$, $Z_{b_k}$, $Z_{x_k}$ to be zero identifies

$$\nabla_v \rho = Z_v := \left( \frac{\lambda_0^\top x_1}{v^\top v} x_1^\top - \lambda_0^\top \right), \quad \nabla_A \rho = Z_A := \left[ \sum_{k=1}^K x_k \lambda_k^\top \right]. \tag{31}$$

Theorem 4.2 is complete.

## D  Proof of Corollary 4.3

The following derivation covers only the case for Lanczos, i.e., we use the variables $x_1, ..., x_{K+1}$ instead of Arnoldi's $q_1, ..., q_K$. But the derivation is the same for both methods.

The expression we manipulate is

$$d\rho = \operatorname{trace}\left( \left[ \sum_{k=1}^K x_k \lambda_k^\top \right] dA \right) + \operatorname{const} \tag{32}$$

where all non-$dA$-related quantities are treated as some unimportant constants.

For a single parameter $\theta_j$, $dA = (\nabla_{\theta_j} A)^\top d\theta_j$ and we have

$$\operatorname{trace}\left( \left[ \sum_{k=1}^K x_k \lambda_k^\top \right] D_j A d\theta_j \right) = \operatorname{trace}\left( \left[ \sum_{k=1}^K x_k \lambda_k^\top \right] (\nabla_{\theta_j} A)^\top \right) d\theta_j \qquad \text{(chain rule)}$$

$$= \operatorname{trace}\left( \sum_{k=1}^K \lambda_k^\top (\nabla_{\theta_j} A)^\top x_k \right) d\theta_j \quad \text{(cyclic property of traces)}$$

$$= \sum_{k=1}^K \lambda_k^\top (\nabla_{\theta_j} A)^\top x_k d\theta_j \qquad \text{(a scalar is its own trace)}$$

$$= \nabla_{\theta_j} \left[ \sum_{k=1}^K \lambda_k^\top A^\top x_k \right] d\theta_j \qquad \text{(linearity of diff. \& summation)}$$

$$= \nabla_{\theta_j} \left[ \sum_{k=1}^K \lambda_k^\top A x_k \right] d\theta_j. \qquad \text{(symmetry of } A\text{)}$$

In conclusion, the derivative of $\rho$ wrt $\theta_j$ is

$$\nabla_{\theta_j} \rho = \nabla_j \left[ \sum_{k=1}^K x_k^\top A^\top \lambda_k \right] \tag{33}$$

and stacking all of those partial derivatives on top of each other, we obtain

$$\nabla_\theta \rho = [\nabla_{\theta_j} \rho]_j = \nabla \left[ \theta \mapsto \sum_{k=1}^K x_k^\top A(\theta)^\top \lambda_k \right] = \sum_{k=1}^K \nabla \left[ \theta \mapsto A(\theta)^\top \lambda_k \right] x_k. \tag{34}$$

We already compute $A(\theta)^\top \lambda_k$ during the backward pass, so we are a single vector-Jacobian product with $x_k$ away from a matrix-parameter-gradient instead of a matrix-gradient. This requires $O(p+n)$ storage, and is computed online, which makes the memory-complexity independent of $K$.

## E  Solving the adjoint system

The upcoming section details how to solve the adjoint system for both, the Lanczos and the Arnoldi iterations. It reuses notation from Appendices B and C.

We begin with Lanczos, because the solution is less technical, and because starting with Lanczos can provide a template for solving Arnoldi's adjoint system. All results in the present section (except for those that explicitly point to Deuflhard et al. [93], Deuflhard [94], which are marked as such) are new and a contribution of this work.

### E.1 Lanczos

The inputs to the adjoint system are the Lanczos vectors $\{x_k\}_{k=1}^{K+1}$ and the coefficients $\{a_k\}_{k=1}^{K}$ as well as $\{a_k\}_{k=1}^{K}$ from the forward pass, the corresponding input derivatives $\{\nabla_{x_k}\rho\}_{k=1}^{K+1}$, $\{\nabla_{a_k}\rho\}_{k=1}^{K}$, and $\{\nabla_{a_k}\rho\}_{k=1}^{K}$, and matrix $A$ and initial vector $v$.

The overall strategy for solving the adjoint system of the Lanczos iteration (Theorem 4.2) is the following: for every $k = K, ..., 1$, alternate the two steps:

1. Combine orthogonality with the $Z_{a_k}$ constraints to get $\nu_k$, and combine it with the $Z_{b_k}$ constraints to get $\mu_k$.

2. Once each $\mu_k$ and $\nu_k$ are available, solve for $\lambda_k$ and repeat with the next lower $k$.

This results in the following procedure: To start, set $\zeta_{K+1} = -(\nabla_{x_{K+1}}\rho)$ and $\lambda_{K+1} = 0$. Then, for all $k = K, ..., 1$, compute

$$\xi_k = \zeta_{k+1}/b_k \tag{35a}$$

$$\tilde{\mu}_k = \nabla_{b_k}\rho - \lambda_{k+1}^\top x_k + x_{k+1}^\top \xi_k \tag{35b}$$

$$\tilde{\nu}_k = \nabla_{a_k}\rho + x_k^\top \xi_k \tag{35c}$$

$$\lambda_k = -\xi_k + \tilde{\mu}_k \cdot x_{k+1} + \tilde{\nu}_k \cdot x_k \tag{35d}$$

$$\zeta_k = -\nabla_{x_k}\rho - A^\top \lambda_k + a_k \cdot \lambda_k + b_k \cdot \lambda_{k+1} - b_k \cdot \tilde{\nu}_k \cdot x_{k+1} \tag{35e}$$

$$\text{Repeat with } k = k - 1. \tag{35f}$$

Finally, set $\lambda_0 = \zeta_1$. Only $\lambda$ and $\zeta$ affect subsequent steps; $\mu$, $\nu$, and $\xi$ are only needed for computing $\lambda$ and $\zeta$. The $k$-th step depends on $a_k, b_k, x_k, x_{k+1}, \nabla_{a_k}\rho, \nabla_{b_k}\rho, \nabla_{x_k}\rho$.

The strategy above yields all $\{\lambda_k\}_{k=0}^{K}$. Finalise the gradients

$$\nabla_v \rho = \frac{\lambda_0^\top x_1}{v^\top v} x_1 - \lambda_0, \quad \nabla_A \rho = \sum_{k=1}^{K} \lambda_k x_k^\top \tag{36}$$

which can be embedded into any reverse-mode algorithmic-differentiation-engine.

### E.2 Arnoldi

The process for Arnoldi is similar to that for Lanczos, but the derivation is more technical. It shares many similarities with deriving the Arnoldi iteration (i.e., the forward pass), so we begin by providing a perspective on recurrence relations (which include the Arnoldi iteration) through the lens of linear system solvers [93, 94] before we use this perspective to solve the adjoint system.

#### E.2.1 Solving the original system

At its core, finding the Arnoldi vectors amounts to solving

$$-AQ + QH + r(e_K)^\top = 0, \tag{37a}$$

$$Qe_1 - cv = 0, \tag{37b}$$

which is possible in closed form as follows: We rewrite the first two constraints as

$$(e_1 \otimes I)\text{vec}\,(Q) = \text{vec}\,(cv) \tag{38a}$$

$$-(I \otimes A)\text{vec}\,(Q) + (H^\top \otimes I)\text{vec}\,(Q) + (e_K \otimes I)\text{vec}\,(r) = 0. \tag{38b}$$

This expression is equivalent to

$$\begin{pmatrix} e_1 \otimes I & 0 \\ H^\top \otimes I - I \otimes A & e_K \otimes I \end{pmatrix} \begin{pmatrix} \text{vec}\,(Q) \\ \text{vec}\,(r) \end{pmatrix} = \begin{pmatrix} \text{vec}\,(cv) \\ 0 \end{pmatrix} \in \mathbb{R}^{N(K+1)\times 1} \tag{39}$$

This is similar to the work by Deuflhard [94], who explain adjoints of three-term recurrence relations. Since $c^2 = \langle v, v \rangle$ holds (ie, $c$ is known), the first row of $Q$ is known. Then, the first row of $Q$ together with the orthogonality constraints yields the first row of $H^\top$, which then defines the next row of the linear system thus the second row of $Q$. Alternating between deriving the next row of $H^\top$ and solving the lower triangular systems is then Arnoldi's algorithm:

**Algorithm E.1** (Arnoldi's forward pass; paraphrased). Assume that $v$ and $K$ are known. Compute $c = \sqrt{\langle v, v \rangle}$. Then, for $k = 1, ..., K$, alternate the following tsteps:

1. Derive the next column of $H$ using the orthogonality constraints.

2. Forward-substitute ("solve") the block-lower-triangular system for the next column of $Q$ (respectively $r$ at the last iteration).

Return all $Q$ and $H$, as well as $c$ and $r$.

The same principle applies to the adjoint system, and the only difference is that the notation is slightly more complicated:

### E.2.2 Solving the adjoint system

The constraints $Z_Q = 0$ and $Z_r = 0$ mirror those of $AQ - QH - r(e_K)^\top = 0$ and $Qe_1 - cv = 0$; the constraints $Z_H = 0$ and $Z_c = 0$ mirror the orthogonality constraints $Q^\top Q = I$ and $Q^\top r = 0$. Therefore, we start with $Z_Q = 0$ and $Z_r = 0$,

$$\nabla_Q f + A^\top \Lambda - \Lambda H^\top + \lambda(e_1)^\top + Q(I_\leq \circ \Gamma) + Q(I_\leq \circ \Gamma)^\top + r\gamma^\top = 0 \qquad (40\text{a})$$

$$\nabla_r f - \Lambda e_K + Q\gamma = 0. \qquad (40\text{b})$$

Introduce the auxiliary quantities

$$\Psi(\Gamma, \gamma) := \nabla_Q f + Q(I_\leq \circ \Gamma) + Q(I_\leq \circ \Gamma)^\top + r\gamma^\top \qquad \in \mathbb{R}^{N \times K} \qquad (41\text{a})$$

$$\psi(\gamma) := \nabla_r f + Q\gamma \qquad \in \mathbb{R}^{N \times 1}. \qquad (41\text{b})$$

$\Psi$ and $\psi$ only serve the purpose of simplifying the notation in the coming part; there is no "meaning" associated with them. Vectorise both expressions,

$$(I \otimes A^\top)\text{vec}(\Lambda) - [(H^\top)^\top \otimes I]\text{vec}(\Lambda) + (e_1 \otimes I)\text{vec}(\lambda) = -\text{vec}(\Psi(\Gamma, \gamma)) \qquad (42\text{a})$$

$$[(e_K)^\top \otimes I]\text{vec}(\Lambda) = \text{vec}(\psi(\gamma)) \qquad (42\text{b})$$

and observe that this can be written as a linear system

$$\begin{pmatrix} e_1 \otimes I & I \otimes A^\top - (H^\top)^\top \otimes I \\ 0 & (e_K)^\top \otimes I \end{pmatrix} \begin{pmatrix} \text{vec}(\lambda) \\ \text{vec}(\Lambda) \end{pmatrix} = \begin{pmatrix} -\text{vec}(\Psi(\Gamma, \gamma)) \\ \text{vec}(\psi(\gamma)) \end{pmatrix} \qquad (43)$$

with a system matrix that is the transpose of the system matrix of the forward pass. The matrix is upper triangular, and the equation can be solved with backward substitution provided $\psi(\gamma)$ and $\Psi(\Gamma, \gamma)$ are known.

The defining quantities $\Psi$ and $\psi$ emerge by combining the adjoint recursion with the projection constraints $Z_H = 0$ (for $\Gamma$, which yields $\Psi$) and $Z_r = 0$ (for $\gamma$, which yields $\psi$). We use $Z_c = 0$ to get a single element in $\Gamma$; more on this below. Summarise the adjoint pass:

**Algorithm E.2** (Arnoldi's adjoint pass; paraphrased). Assume $Q$, $H$, $c$, and $r$ as well as the gradients of $f$ with respect to those quantities. Then, compute $\psi$ via computing $\gamma$ using $Z_r = 0$. Then, for $k = K, ..., 1$, alternate the following two steps:

1. Derive the next row of $\Psi$ by combining $Z_Q = 0$ with the projection constraint $Z_H = 0$

2. Backward-substitute ("solve") for the next row of $\Lambda$ (recall: we loop backwards)

Finally, use $Z_c = 0$ to get the first row of $\Psi$ and solve for $\lambda$. Then, return $\nabla_A f = \Lambda Q^\top$ and $\nabla_v f = \lambda c^\top$.

The structure of the adjoint pass is similar to the forward pass (Table 6). In the following, we will elaborate on each of those steps. We assume that the reader knows how to solve a lower triangular linear system. We focus on constructing $\psi$ and $\Psi$ via $\Gamma$ and $\gamma$.

Table 6: Forward versus adjoint (backward) pass

|  | System matrix | Solve via | Recursively define | Using |
|---|---|---|---|---|
| Forward | Lower triangular | Forward substitution | System matrix | Orthogonality |
| Adjoint | Upper triangular | Backward substitution | Right-hand side | Projection: $Z_H = 0$ |

### E.2.3 Initialisation

Initialisation of the adjoint pass implies computing $\psi$. To get $\psi$, we need $\gamma$: Consider multiplying $Z_r$ with $Q^\top$,

$$
\begin{aligned}
0 = Q^\top Z_r &= Q^\top \nabla_r f - Q^\top \Lambda e_K + \gamma && \text{(use the definition of } Z_r\text{)} \\
&= Q^\top \nabla_r f - (\nabla_H f + I_\ll \circ \Sigma) e_K + \gamma && \text{(since } Z_H = 0\text{)} \\
&= (Q^\top \nabla_r f - \nabla_H f e_K) + (I_\ll \circ \Sigma) e_K + \gamma && \text{(reorder)} \\
&= (Q^\top \nabla_r f - \nabla_H f e_K) + \gamma && ((I_\ll \circ \Sigma) e_K = 0)
\end{aligned}
$$

where we use that the last column in $I_\ll \circ \Sigma$ consists entirely of zeros. All other quantities are known. Therefore, $\gamma$ is isolated and we identify

$$
\gamma = \nabla_H f e_K - Q^\top \nabla_r f. \tag{44}
$$

Next, use this $\gamma$ to build $\psi$,

$$
\psi(\gamma) = \nabla_r f + Q\gamma \tag{45}
$$

and the initialisation step is complete.

### E.2.4 Recursion

With $\psi$ in place, we get the last column of $\Lambda$. To get the next column of $\Lambda$, we need to derive the right-hand side $\Psi$. To get $\Psi$, we need $\Gamma$.

Multiply $Q^\top Z_Q$ to obtain

$$
0 = Q^\top Z_Q \tag{46}
$$
$$
= Q^\top \nabla_Q f + Q^\top A^\top \Lambda - Q^\top \Lambda H^\top + Q^\top \lambda (e_1)^\top + I_\le \circ \Gamma + (I_\le \circ \Gamma)^\top
$$
$$
\text{(since } Q^\top Q = I \text{ and } Q^\top r = 0)
$$
$$
= Q^\top \nabla_Q f + Q^\top A^\top \Lambda - (\nabla_H f + I_\ll \circ \Sigma) H^\top + Q^\top \lambda (e_1)^\top + I_\le \circ \Gamma + (I_\le \circ \Gamma)^\top
$$
$$
\text{(since } Z_H = 0)
$$
$$
= (Q^\top \nabla_Q f - \nabla_H f H^\top) + Q^\top A^\top \Lambda + Q^\top \lambda (e_1)^\top + I_\le \circ \Gamma + (I_\le \circ \Gamma)^\top + (I_\ll \circ \Sigma) H^\top.
$$
$$
\text{(reorder the terms)}
$$

Since $H$ is Hessenberg, $(I_\ll \circ \Sigma) H^\top$ is strictly lower triangular. Therefore, multiplication with $I_\ge$ removes $\Sigma$ from the expression,

$$
0 = I_\ge \circ \left[ Q^\top \nabla_Q f - \nabla_H f H^\top + Q^\top A^\top \Lambda + Q^\top \lambda (e_1)^\top + I_\le \circ \Gamma + (I_\le \circ \Gamma)^\top \right] \tag{47a}
$$
$$
= I_\ge \circ \left[ Q^\top \nabla_Q f - \nabla_H f H^\top + Q^\top A^\top \Lambda \right] + I_\ge \circ \left[ Q^\top \lambda (e_1)^\top \right] + I_= \circ \Gamma + I_\ge \circ \Gamma^\top. \tag{47b}
$$

The term involving $\lambda$ can be simplified as follows: Due to the presence of $(e_1)^\top$, we know that $Q^\top \lambda (e_1)^\top$ is lower triangular (in fact, it has a single nonzero column). Thus, $I_\ge \circ (Q^\top \lambda (e_1)^\top)$ is proportional to $e_1 (e_1)^\top$,

$$
\begin{aligned}
I_\ge \circ (Q^\top \lambda (e_1)^\top) &= \left[ (Q e_1)^\top \lambda \right] e_1 (e_1)^\top && \tag{48} \\
&= c v^\top \lambda\, e_1 (e_1)^\top && \text{(since } Q e_1 = c v) \\
&= c \nabla_c f\, e_1 (e_1)^\top && \text{(since } Z_c = 0)
\end{aligned}
$$

and all quantities are known; hence,

$$
I_= \circ \Gamma + I_\ge \circ \Gamma^\top = -I_\ge \circ [Q^\top \nabla_Q f - \nabla_H f H^\top + Q^\top A^\top \Lambda] - c \nabla_c f\, e_1 (e_1)^\top \tag{49}
$$

**Algorithm E.3** (Forward pass). Initialise $k = 1, Qe_1$. Then, for $k = 1, ..., K$:

1. Use orthogonality for a new row in the system matrix in Equation 39.
2. Solve for the next column of $Q$
3. Optional: re-enforce $Q^\top Q = I$.

Solve for $r$ and return $Q, H, r, c$.

**Algorithm E.4** (Backward pass). Initialise $k = K, \Lambda e_1$. Then, for $k = K, ..., 1$:

1. Use projection for a new column of the right-hand side $\Psi$
2. Solve for the next column of $\Lambda$
3. Optional: re-enforce $Z_H = 0$.

Solve for $\lambda$ and return $\nabla_\theta \rho$ and $\nabla_v \rho$.

Figure 8: Forward and backward pass of the Arnoldi iteration (paraphrased)

must hold.

Now, the most important observation is the following: the last column of $I_= \circ \Gamma + I_\geq \circ \Gamma^\top$ depends on the last column of $Q^\top A^\top \Lambda$ and known quantities; the penultimate column depends on the penultimate column of $\Gamma$, and so on. But at the time of assembling the last column of $\Gamma$, the last column of $\Lambda$ is known! More generally, we always know one more column of $\Lambda$ than of $\Gamma$, so we can recursively assemble $I_= \circ \Gamma + I_\geq \circ \Gamma^\top$:

Let

$$\text{sym}(M) := I_\geq \circ M + (I_> \circ M)^\top \tag{50}$$

be a symmetrisation operator. We define it for the sole purpose of reconstructing

$$\text{sym}\left(I_= \circ \Gamma + I_\geq \circ \Gamma^\top\right) = I_\leq \circ \Gamma + (I_\leq \circ \Gamma)^\top. \tag{51}$$

Let us use it:

$$I_\leq \circ \Gamma + (I_\leq \circ \Gamma)^\top = \text{sym}\left(-I_\geq \circ [Q^\top \nabla_Q f - \nabla_H f H^\top + Q^\top A^\top \Lambda] - c\,\nabla_c f\,e_1(e_1)^\top\right) \tag{52a}$$

$$= \text{sym}\left(-I_\geq \circ [Q^\top \nabla_Q f - \nabla_H f H^\top + Q^\top A^\top \Lambda]\right) - c\,\nabla_c f\,e_1(e_1)^\top. \tag{52b}$$

This yields the next row/column of $\Gamma + \Gamma^\top$, and therefore the next row of $\Psi$. From there, we can assemble the next column of $\Lambda$ and iterate. Figure 8 (respectively Algorithms E.3 and E.4) compare pseudocode for forward and adjoint passes. Altogether, the implementation of the adjoint pass is very similar to that of the forward pass.

At the final step, we obtain not the last column of $\Lambda$ but $\lambda$, though this is a byproduct of solving the triangular linear system. It does not need further explanation.

**Remark E.5** ($\Sigma$). *Like for gradients of QR decompositions [61, 75], we never solve for $\Sigma$.*

## F   Setup for Table 2

To create Table 2, we implement an operator

$$\mathcal{I} : A \mapsto (H, Q, r, c) \mapsto QHQ^\top \tag{53}$$

where $(H, Q, r, c)$ are the result of a full-rank Arnoldi iteration (i.e. $K = N$). For $K = N$, $QHQ^\top = A$ and $\mathcal{I}$ must have an identity Jacobian; thus,

$$\varepsilon := \|I_{N^2} - \mathcal{I}\|_{\text{RMSE}} \tag{54}$$

measures the loss of accuracy when differentiating the Arnoldi iteration. A small $\varepsilon$ is desirable.

Then, using double-precision, we construct a Hilbert matrix $A = [1/(i + j + 1)]_{i,j=1}^N \in \mathbb{R}^{N \times N}$ which is a famously ill-conditioned matrix and a common test-bed for

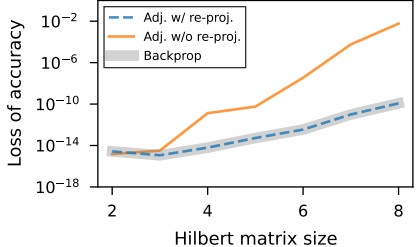

Figure 9: Accuracy loss $\varepsilon$ when differentiating $\mathcal{I}$ for a Hilbert matrix of increasing size $N$. Uses double precision.

the loss of orthogonality in methods like the Lanczos and Arnoldi iteration [e.g. 71, Table 7.1]. We evaluate three algorithms, all of which rely on the Arnoldi iteration with full reorthogonalisation on the forward-pass: One algorithm does not re-project on the adjoint constraints, another one does, and

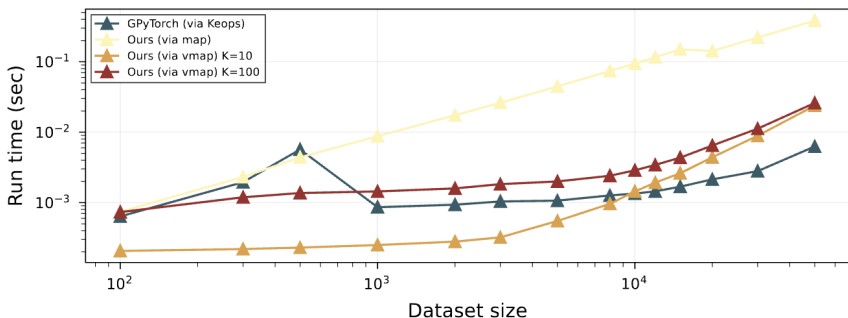

Figure 10: For matrices with at least 10,000 rows/columns, KeOps remains the state of the art. This experiment uses a square-exponential kernel, on an artificial dataset with $d = 3$ dimensions.

for reference we compute $\varepsilon$ when "backpropagating through" the re-orthogonalised Arnoldi iteration as a third option. Figure 3 has demonstrated that the first two options beat the third one in terms of speed, but we consider numerical accuracy here.

We evaluate $\varepsilon$ for $N = 1, ..., 8$ (see Figure 9), and show the values for $N = 8$ in Table 2. The numerical accuracy of the re-projected adjoint method matches that of differentiating "through" re-orthogonalisation, and outperforms not re-projecting by a margin.

## G   Memory-efficient kernel-matrix-vector products in JAX

Matrix-free linear algebra requires efficient matrix-vector products. For kernel function $k = k(x, x')$, and input data $x_1, ..., x_N$, Gaussian process covariance matrices are of the form $A = [k(x_i, x_j)]_{i,j=1}^N$. Matrix-vector products with $A$ thus look like

$$v \mapsto Av = \left[ \sum_{j=1}^N k(x_i, x_j) v_j \right]_{i=1}^N \tag{55}$$

and can be assembled row-wise, either sequentially or parallely.

The more rows we assemble in parallel, the faster the runtime but also the higher the memory requirements, so we follow Gardner et al. [6] and choose the largest number of rows of $A$ that still fit into memory, say $r$ such rows, and assemble $Av$ in blocks of $r$. In practice, we implement this in JAX by combining `jax.lax.map` and `jax.vmap`, but care has to be taken with reverse-mode automatic differentiation through $(v, \theta) \mapsto A(\theta)v$ because by default, reverse-mode differentiation stores all intermediate results. To solve this problem, we place checkpoints around each such batch of rows, which reduces the memory requirements but roughly doubles the runtime. (We place another checkpoint around each stochastic trace-estimation sample, which roughly doubles the runtime again.)

An alternative to doing this manually is the KeOps library [82], which GPyTorch [6] builds on. However, there currently exists no JAX-compatible interface to KeOps which is why we have to implement the above solution.

Figure 10 compares the runtime of our approach to that of KeOps custom CUDA code. We see that we are competitive, but roughly 5× slower for medium to large datasets. Multiplying this with the 4× increase due to the checkpoints discussed above explains the 20× increase in runtime compared to GPyTorch. Being 20× slower than GPyTorch per epoch is only due to the matrix-vector products, and has nothing to do with the algorithm contribution. Future work should explore closing this gap with a KeOps-to-JAX interface.

## H   Experiment configurations for the Gaussian process study

**Data**   For the experiments we use the "Protein", "KEGG (undirected", "KEGG (directed)", "Elevators", and "Kin40k" datasets (Table 7, adapted from Bartels et al. [95]). All are part of the UCI data

Table 7: Datasets used in this study.

| Dataset | Source |
|---|---|
| Protein | Available here.[3] |
| Elevators | Camachol [96] |
| Kin40K | Schwaighofer and Tresp [97] |
| KEGG (undir) | Shannon et al. [98] |
| KEGG (dir) | Shannon et al. [98] |

repository, and accessible through there.

The data is subsampled to admit the train/test split of 80/20%, and to admit an even division into the number of row partitions. More specifically, we use 10 partitions for the kernel-matrix vector products. This way, we have to discard less than 1% of the data; e.g., on KEGG (undir), we use 63,600 instead of the original 63,608 points.

We calibrate a Matérn prior with smoothness $\nu = 1.5$, using 10 matrix-vector products per Lanczos iteration, conjugate gradients tolerance of $\epsilon = 1$, a rank-15 pivoted Cholesky preconditioner, and 10 Rademacher samples. We evaluate all samples sequentially (rematerialising on the backward pass to save memory, as discussed in Appendix G). The conjugate-gradients tolerances are taken to be absolute (instead of relative), and the parametrisations of the Gaussian process models and loss functions match that of GPyTorch.

For every model, we calibrate an independent lengthscale for each input dimension, as well as an scalar observation noise, scalar output-scale, and the value of a constant prior mean. All parameters are initialised randomly. We use the Adam optimiser with learning rate 0.05 for 75 epochs. All experiments are repeated for three different seeds.

# I   Partial differential equation data

We generate data for the differential equations as follows: Recall the problem setup of a partial differential equation

$$\frac{\partial^2}{\partial^2 t} u(t; x_1, x_2) = \omega(x_1, x_2)^2 \left( \frac{\partial^2}{\partial x_1^2} u(t; x_1, x_2) + \frac{\partial^2}{\partial x_2^2} u(t; x_1, x_2) \right) \tag{56}$$

with Neumann boundary conditions. The coefficient field $\omega$ is space- but not time-dependent.

First, we discretise the Laplacian operator with central differences on an equidistant, tensor-product mesh that consists of 128 points per dimension, which yields $128^2$ grid points. The resulting second-order ordinary differential equation

$$\frac{\mathrm{d}^2}{\mathrm{d}t^2} w = \omega^2 M w, \tag{57}$$

where $M$ is the discretised Laplacian, is then transformed into a first-order differential equation

$$\frac{\mathrm{d}}{\mathrm{d}t} \begin{pmatrix} w \\ \dot{w} \end{pmatrix} = \begin{pmatrix} 0 & I \\ \omega^2 M & 0 \end{pmatrix} w =: Aw. \tag{58}$$

This equation is solved by the matrix exponential, and the system matrix $A$ is asymmtric (by construction), and highly sparse because $M$ is. Matrix-vector products with $A$ are cheap, because we can implement them with `jax.scipy.signal.convolve2d`.

Then, we sample a true $\omega$ from a Gaussian process with a square-exponential covariance kernel, using lengthscale `softplus(−0.75)` and output-scale `softplus(−10)`. We sample from this process with the Lanczos algorithm [11] using Krylov-depth $K = 32$.

Then, we use another Gaussian process with the same kernel, but lengthscale `softplus(0)` and output scale `softplus(0)`, to sample 256 initial distributions – again with the Lanczos algorithm

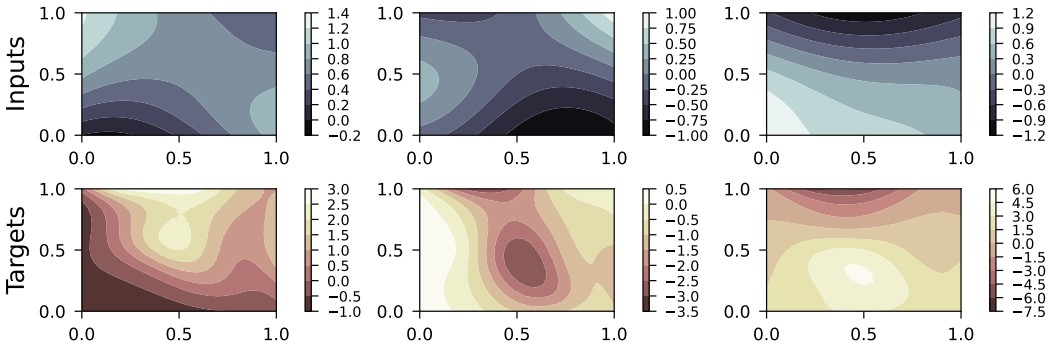

Figure 11: Three exemplary input/output pairs from the PDE dataset.

[11]. These 256 initial conditions are solved with Diffrax's implementation of Dopri8 [99] using 128 timesteps. Some example input/output pairs are in Figure 11.

This setup is similar to that of the WaveBench dataset [92], with the main difference being that the WaveBench dataset uses a slightly different formulation of the wave equation.[4] We use the one above because it lends itself more naturally to matrix exponentials, which are at the heart of this experiment.

## J    Implementation details for the Bayesian neural network study

### J.1    Bayesian neural networks with Laplace approximations

Another possible application of the gradients of matrix functions is marginal-likelihood-based optimisation of Bayesian Neural Networks. Suppose $g_\theta(x)$ is the output of a neural network with parameters $\theta \in \mathbb{R}^P$. The choice of the model shall be denoted by $\mathcal{M}$ and consist of both continuous and discrete hyperparameters (such as network architecture, likelihood precision, prior precision, etc.). For some choice of prior given by

$$p(\theta \mid \mathcal{M}) \tag{59}$$

and likelihood

$$p(y|x, \theta, , \mathcal{M}) = p(y \mid x, g_\theta(x), \mathcal{M}) \tag{60}$$

we can specify a Bayesian model. The posterior distribution is then given by:

$$p(\theta, y \mid x, \mathcal{M}) \propto p(y \mid x, g_\theta(x), \mathcal{M})p(\theta \mid \mathcal{M})d\theta. \tag{61}$$

The marginal likelihood is given by normalizing constant of this posterior, i.e.

$$p(y \mid x, \mathcal{M}) = \int p(y \mid x, g_\theta(x), \mathcal{M})p(\theta \mid \mathcal{M})d\theta. \tag{62}$$

As suggested by MacKay [101], this marginal likelihood can be used for model selection in Bayesian neural networks. Immer et al. [30] use the Laplace approximation of the posterior to obtain access to the marginal likelihood of the Bayesian neural network and its stochastic gradients.

The Laplace approximation of the marginal likelihood is given by:

$$\log p(y \mid x, \mathcal{M}) \approx \log p(y, \theta_{\mathrm{MAP}} \mid x, \mathcal{M}) - \frac{1}{2}\log\det\left(\frac{1}{2\pi}\boldsymbol{H}_{\theta_{\mathrm{MAP}}}\right) \tag{63}$$

where $\boldsymbol{H}_{\theta_{\mathrm{MAP}}} = -\nabla^2_\theta \log p(y, \theta_{\mathrm{MAP}} \mid x, \mathcal{M})$. Usual choices of the prior are $N(0, \alpha^{-1}\mathbb{I})$. Usually this Hessian is approximated with the *generalized Gauss-Newton* (GGN) matrix [102]

$$\boldsymbol{H}_{\theta_{\mathrm{MAP}}} \approx A(\alpha) \coloneqq \sum_{j=1}^{J}[D_\theta g_{\theta_{\mathrm{MAP}}}])(x_j)^\top [D_g^2\rho](y_j, g_{\theta_{\mathrm{MAP}}}(x_j))[D_\theta g_{\theta_{\mathrm{MAP}}}])(x_j)^\top + \alpha^2 I \tag{64}$$

---

[3]Link: http://archive.ics.uci.edu/dataset/265/physicochemical+properties+of+protein+tertiary+structure

[4]To ensure radiating boundary conditions, Liu et al. [92] follow Stanziola et al. [100]'s model of simulating the wave equations as a sytem of first-order equations.

where $D^2\rho$ is the Hessian of the loss, and $D_\theta g$ the Jacobian of $g$ (recall Equation 2). This objective is used to optimize the prior precision of the model or any continuous model hyperparameters. Matrix-vector products with the GGN matrix can be accessed through automatic differentiation using Jacobian-vector and vector-Jacobian products. With these efficient matrix-vector products, one can estimate the log-determinant of GGN using matrix-free techniques like the Lanczos iteration.

To make predictions using the Laplace approximation of the posterior, we also need to sample from the normal distribution $N(\theta_{\mathrm{MAP}}, A^{-1})$. Samples from this distribution can be written as:

$$\theta = \theta_{\mathrm{MAP}} + A^{-1/2}\epsilon \tag{65}$$

where $\epsilon \sim N(0, I)$. The main bottleneck in this computation is the inversion and matrix square root of the GGN matrix, and we implement it with a Lanczos iteration using $f(x) = x^{-1/2}$. Since the GGN is empirically known to have low-rank [103], doing a few Lanczos iterations can get us close to an accurate estimation.

## J.2 Experiment setup

We estimate the diagonal of the GGN stochastically via ("$\circ$" is the element-wise product) [104]

$$\mathrm{diagonal}\,(A) = \mathbb{E}[v \circ Av] \approx \frac{1}{L}\sum_{\ell=1}^{L} v_\ell \circ Av, \quad \mathbb{E}[vv^\top] = I. \tag{66}$$

We use 150 matrix-vector products for both diagonal calibration and our Lanczos-based estimation. We use 30 Monte-Carlo samples to estimate the log-likelihoods for evaluating the test metrics, and we use `places365` [91] as an out-of-distribution dataset to compute OOD-AUROC. We also compute the expected calibration error (ECE) [105] of the model.

**Data:** We show scalability by doing Laplace approximation on Imagenet1k image classification [89]. The training set consists of approximately 1.2 million images, each belonging to one of 1000 classes. We find that we can take small subsets of this dataset and still converge to the same prior precision. Our computational budget allows us to use 10 percent of the samples for each class. However, even for very small subsamples of the data, we converge to a very similar prior precision.

**Method:** To optimize the prior precision we use the marginal likelihood as the objective. We use the RMSprop optimizer with a learning rate of 0.01 for 100 epochs for optimizing both the diagonal and Lanczos approximations of the GGN.

