# OpenReview forum: "Gradients of Functions of Large Matrices"
_NeurIPS.cc/2024/Conference — NeurIPS 2024 spotlight_

### Official Review · Reviewer_FpA9 · 2024-06-29

**Soundness:** 3
**Presentation:** 4
**Contribution:** 3
**Rating:** 7
**Confidence:** 3

**Summary:**

This paper introduces a new matrix free method for automatically differentiating functions of matrices. The computational problem discussed in this paper is of interest because the matrix dimension scales with respect to the size of the dataset (e.g. Gaussian process regression, etc.). The authors' algorithm yields the exact gradients of the forward pass, all gradients are obtained with the same code, and said code runs in linear time- and memory-complexity. The proposed method is also matrix-free and does not form the matrix explicitly which is the key to scalability. The authors' method evaluate functions of large matrices by: 1) first decomposing the matrix into a product of small matrices (e.g. Lanczos, Arnoldi iterations); 2) then evaluating functions on the small matrix produced during the Lanczos/Arnoldi iteration. The authors note that functions of small matrices can be differentiated efficiently. The authors provide the method to compute gradients of the Lanczos/Arnoldi iteration so that backpropagation can be implemented efficiently. The authors use differentiation via adjoints of the Lanczos/Anroldi iterations (this is described in Theorem 4.1 and Theorem 4.2). The parameter gradient expression that can be used for any problem is provided in Corollary 4.3. The authors provide experimental results which provide impressive speedup over GPyTorch over problems of interest including: Gaussian process, physics-informed PDEs, Bayesian neural network tuning.

**Strengths:**

1. This paper tackles an important computational bottleneck in tuning hyperparameters of potentially complex models that are used in scientific and machine learning applications.
2. The proposed method is general and can accomodate any matrix functionals and do not need explicit derivations of the gradients.
3. The authors provide experimental results on several problems of interest to machine learning practitioners.

**Weaknesses:**

There are no particular weaknesses I found on reading the paper. The paper is well written.

**Questions:**

1. There are parallelization techniques that may be useful for even scaling up the proposed method.
2. How many samples do you take for stochastic estimation of log-determinant for the exact Gaussian process result?
3. What norm is the gradient error measured in?

**Limitations:**

The authors describe several limitations - but as the authors mention, these extensions describe a more expanded write-up.

---

> ### Author Rebuttal · Authors · 2024-08-05
>
> Thank you so much for your positive evaluation!
>
> To answer your questions:
>
> 1. We agree that these directions are interesting for future research!
> 2. We use 10 Rademacher samples to match GPyTorch's default settings. Appendix H explains all Gaussian-process-related parameters.
> 3. The errors are relative mean-squared errors (using the square root of the machine epsilon in the current floating-point accuracy as a nugget to avoid dividing by zero). We accidentally omitted this information in the submission and will add it to the next version.
>
> Thanks again for your review!
> We hope that you continue to fight for this paper's acceptance.

---

### Official Review · Reviewer_X6aJ · 2024-07-11

**Soundness:** 3
**Presentation:** 4
**Contribution:** 3
**Rating:** 8
**Confidence:** 4

**Summary:**

There are some useful iterative methods for calculating important matrix products, to wit, Lanczos and Arnoldi iterations, which apparently did not have known derivatives, until now. The paper provides a generic framework for calculating the derivatives of such iterative linear operator approximations. The core of the paper is interpreting the Lanczos resp Arnoldi iteration as an iterative system and using the implicit derivative trick

**Strengths:**

While these results are not *world* shaking, they seem to provide immediate quality-of-life improvements to linear algebra users in high-value problems. The results unify several known "matrix free" tricks, which is satisfying, and point the way to more.

The paper is well-written, compact, and easy to understand, which is a great relief this deep into the review process.

**Weaknesses:**

As far as I can see, few. The paper seems to present what I need to know.

It would not be impossible for these results to be known elsewhere in the literature, but if they are known, I have not seen them.

**Questions:**

I have no questions. This paper was well explained, and clear about its goals, and how it approached them.

**Limitations:**

There are many limitations to the methods divulged here, but they seem to be adequately articulated in the paper.

---

> ### Author Rebuttal · Authors · 2024-08-05
>
> We greatly appreciate your positive assessment!
> We are happy that you acknowledge how our work "provide(s) immediate quality-of-life improvements to linear algebra users in high-value problems" because this was precisely our goal!
> We hope that you continue to fight for this paper's acceptance.

---

> > ### Comment · Reviewer_X6aJ · 2024-08-11
> >
> > My enthusiasm for these results is based on the fact that I needed more-or-less exactly this result for one of my own papers in 2022. I salute the authors for actually *finding* it where I failed.

---

### Official Review · Reviewer_3YyQ · 2024-07-12

**Soundness:** 3
**Presentation:** 3
**Contribution:** 3
**Rating:** 7
**Confidence:** 4

**Summary:**

The paper proposes an adjoint method for functions of matrices that utilize Arnoldi/Lanczos iterations to compute gradients with respect to large dimensional variables and demonstrates their approach's utility on a variety of common compute/memory intensive tasks encountered in machine learning. They derive the adjoints by implicitly differentiating the solution of Arnoldi/Lanczos iterations with respect to the input matrix and vector and then show how this can be used to compute gradients with respect to parameters that the input matrix depends on. They then examine their method on three case studies: Gaussian process hyperparameter optimization, solving for parameters of a discretized two-dimensional (in space) PDE and post-training calibration of a Bayesian neural network. Their results demonstrate an improvement over the standard approaches on a range of metrics.

**Strengths:**

- An adjoint method for the Arnoldi iteration is certainly a useful addition to the automatic differentiation toolbox given the techniques wide use and would thus be of significant interest to the Neurips community.
- The paper was mostly clear to read and understand. The conciseness of the experiment section is also a strength as the key aspects of the setups and the results are succinctly presented.
- The paper's results indicate the methods utility in obtaining better runtimes, more accurate gradients and/or lower train and test losses on a range of important tasks in machine learning. In particular, the BNN calibration experiment clearly demonstrates how their method can handle matrices that may be too large to fit on a GPU and so it doesn't have to resort to subsequent approximation which they show leads to better performance.

**Weaknesses:**

- Although the paper empirically shows how backprop through a matrix function is slower than their adjoint method for Arnoldi iterations in Fig 3, the fact that a sparse matrix (as opposed to a dense one) was utilized motivates some skepticism over whether we would expect to see this same runtime relationship for a dense matrix. In fact, the exact GP results suggest that implementing matrix-matrix (/-vector) operations more efficiently can be enough to make reverse-mode AD work quicker for dense matrices. More discussion on why the adjoint Arnoldi iteration is more appropriate would strengthen their argument for the improved efficiency of their method.
- The description of the adjoint systems in section 4.1 is lacking some explanation to make clear the key aspects of the adjoint methods and why this is preferable over reverse-mode AD. Reverse-mode AD also only utilizes vector-Jacobian products and so is a kind of "matrix-free" method. Explanation of how their method differs from this would make more apparent their contribution.

**Questions:**

- The last part of the explanation of implicit differentiation in section 4 was confusing. On the one hand $\mathrm{d}\rho$ is defined in eqn (9) but the form $\mathrm{d}\rho = \langle \nabla_{\theta} \rho, \mathrm{d}\theta \rangle$ is given on the following line. It was not clear as to why these two should be equal. Being more concrete on the definition of $\rho$ would help here
- As a solution to the second weakness listed above an algorithm comparing adjoint Arnoldi/Lanczos with reverse-mode AD may help to make clear the differences between the methods.
- On the sentence starting with "Gradients ..." on line 198, doesn't reverse-mode AD also only use matrix-vector products as well?
- A definition of "loss of accuracy" in Table 2 should probably be given in the main text.
- A bit of discussion as to why the JAX low-memory matrix-vector product implementations cannot compete with KeOps would be a worthwhile addition to the paper.

**Limitations:**

Limitations on the conceptual aspect of the approach are discussed at the end of section 3 but some discussion in light of the experimental results would also be of value

---

> ### Author Rebuttal · Authors · 2024-08-05
>
> Thanks for the positive evaluation!
>
> Before we answer your questions, we would like to reply to your points listed under "Weaknesses" briefly:
>
> **Matvecs, sparse/dense complexity:**
> The complexity of our adjoint mirrors that of the forward pass of Lanczos/Arnoldi and depends almost entirely on the efficiency of the matrix-vector product. Therefore, Lanczos and Arnoldi are mainly used on sparse or highly structured matrices like PDE discretisations or Jacobians of neural networks, which is why we use a sparse matrix in Figure 3.
> With dense matrices, an efficient matrix-vector product can quickly become the limiting factor already for the forward pass (which is what happens in the Gaussian process case study; see the discussion in Appendix G).
> Finally, unless we misunderstand your sentence
>
>     In fact, the exact GP results suggest that implementing matrix-matrix (/-vector) operations more efficiently can be enough to make reverse-mode AD work quicker for dense matrices
>
> we would like to reinforce that the Gaussian process case study does not compare to 'backpropagation through the solver' but an alternative custom gradient operation (alternative to our adjoint), as implemented in GPyTorch; see Equation 5 and/or the introduction of Section 5.
>
>
>
> **Reverse-mode AD:**
> When you refer to reverse-mode AD in 'why this (the adjoint method) is preferable over reverse-mode AD', do you mean what we call 'backpropagation through the solver'? The adjoint method is one of many ways of implementing reverse-mode AD. If you are looking for a comparison of how backpropagation through the solver relates to our gradient implementation: we show in Figure 3 that our code is far more efficient. See also Appendix A for additional information.
>
>
> **Answers to questions:**
> 1. Thank you for bringing that up. The definition is that $\mathrm{d}x$ is an infinitesimal perturbation. The gradient identity is additional information (and critical for the following derivations). We will update the sentence accordingly.
> 2. We agree that this is crucial; see Figure 3.
> 3. Thank you for bringing that up. If we back-propagate ''through'' Lanczos/Arnoldi, the resulting (automatic) gradients would use reverse-mode derivatives of matrix-vector products. Since our algorithm replaces this step, the sentence in line 198 emphasises that our code is matrix-free, just like the forward pass; see also Corollary 4.3.
> We will revise the sentence to make this distinction more clear.
> 4. Equation 54 in Appendix F formally defines the loss of accuracy. We will link this information more clearly in the main text.
> 5. We agree! We dedicate Appendix G to discussing JAX vs KeOps.
>
>
> Thank you again for your review!

---

> > ### Comment · Reviewer_3YyQ · 2024-08-10
> >
> > I would like to thank the authors for addressing my concerns.
> >
> > So reverse-mode AD does refer to backpropagating through the solver here.
> >
> > I am satisfied with the responses and I will leave my score as is.

---

### Official Review · Reviewer_RkAM · 2024-07-15

**Soundness:** 4
**Presentation:** 4
**Contribution:** 3
**Rating:** 6
**Confidence:** 4

**Summary:**

This paper proposes a new approach to perform automatic differentiation for function of large matrices. Specifically, the paper outlines the backward computation of the matrix-vector product f(A(\theta)) * v where A(\theta) is the jacobian of a large NN that will not fit into memory. The proposed approach uses the Lanczos and Arnoldi iteration method to factorize the matrix-vector product, which permits inexpensive backpropagation, and then derive the backward computation for the Lanczos and Arnoldi iteration via the adjoint method. This technique is tested in three different scenarios that require the evaluation of such matrix-vector product: Gaussian processes (GP), Physics Informed ML with PDE, and Bayesian NN.

**Strengths:**

- Theoretically interesting approach to auto diff of large Jacobian vector product
- Potentially useful in scenarios where functions of large matrices are involved in the objective function
- Interesting case studies detailing several such scenarios and how to apply the proposed method in those scenarios
- Improved performance in majority of experiments
- Generally clear and self-contained presentation, easy to read and follow

**Weaknesses:**

1. Motivation is somewhat unclear. The paper claims that the proposed method provides exact gradient of the forward pass, but it doesn't seem true when the Lanczos & Arnoldi iterations themselves do not yield an exact factorization (Eq. 7). This method, although theoretically quite interesting, should belong to the same class with other approximation methods. As such, I would like the authors to explicitly discuss the benefit of this approximation compared to previous work (which are currently positioned as being inferior to an exact gradient method)

2. Empirical results are not convincing:
- Table 3 seems to highlight the wrong improvement. It is moot to compare training loss when the loss landscapes are different. The real performance measure should be RMSE, in terms of which both methods converge to the same values (although the proposed Arnoldi method is 20x slower -- so I'm not sure what's the benefit here)
- Case study 6 again shows the same performance between the Arnoldi method & Dopri5.
- Bayesian NN is usually trained with Variational Inference. Diagonal method is a rather crude approximation, and it is not surprising that the proposed method outperforms it.

**Questions:**

I have no further question since the method is sound & clearly presented. My only problem is that the empirical results are really limited and do not show any significant improvement over existing methods.

**Limitations:**

No potential negative societal impacts

---

> ### Author Rebuttal · Authors · 2024-08-05
>
> Thanks for the review and the positive assessment!
>
> We would like to briefly reply to the points you list as weaknesses:
>
> **Motivation:**
> You are correct that the Lanczos and Arnoldi iterations yield approximate matrix-function-vector products. When we write "exact gradient of the forward pass", we mean that our adjoints yield the exact gradient of the Lanczos/Arnoldi approximation (the forward pass). We do not claim that it is the exact gradient of the matrix function.
>
> **Table 3:**
> We agree that the RMSE is crucial. However, since neither method beats the other in terms of RMSE, Table 3 does not highlight any RMSE-winners. For context, kegg_dir also has no winner in the training loss. As for the interpretation of this experiment, we consider it a success that our new black-box algorithm matches the performance of a technique that (i) specialises in Gaussian-process log-determinants and (ii) has been honed for many years. We discuss this in the "Analysis" block on page 7. The 20x runtime difference is not due to our algorithm. Instead, it is because the reference code (GPyTorch) relies on hand-crafted CUDA kernels through the KeOps library. The 20x performance difference is due to the inability to use this library in JAX; see the discussion in Appendix G.
>
> **Table 6:**
> Here, too, is the main message that our black-box Arnoldi code can compete with state-of-the-art differential equation adjoints without specialising in ODEs/PDEs. We use the same algorithm for all three case studies, whereas all reference codes specialise in each respective task. For example, the differentiate-then-discretise adjoint for Dopri5 may be fast (as fast as the Arnoldi adjoint for the linear PDE) but only works for ODEs (respectively space-discretised, time-dependent PDEs). Please note how, in Figure A2, our adjoints are more efficient than Dopri5 with a differentiate-then-discretise adjoint in terms of the number of matrix-vector products.
>
> **BNN:**
> We respectfully disagree with the "usually" in your statement that "Bayesian NN is usually trained with Variational Inference".
> There are many strategies for setting up Bayesian neural networks, including the Laplace approximation and variational inference; see Papamarkou et al. (2024).
> However, unlike other methods, the Laplace approximation is particularly relevant for our work because marginal-likelihood calibration requires (the gradient of) a function of a large matrix: the Gauss--Newton matrix.
> Our paper does not demonstrate how Laplace approximations compare to other techniques.
> Instead, it outlines how differentiable linear algebra makes Laplace approximations more efficient.
> Our results show how this has been successful: Our unspecialised implementation of matrix functions beats specialised Laplace-approximation codes for the VAN model. We believe that this reinforces the importance of differentiable linear algebra for advances in probabilistic machine learning.
>
>
> Thank you again for the review; we hope that we were able to clarify a few potential misconceptions. In either case, we are looking forward to your reply!
>
> **Reference:**
> Papamarkou, Theodore, et al. "Position: Bayesian Deep Learning is Needed in the Age of Large-Scale AI." Forty-first International Conference on Machine Learning. 2024.

---

> > ### Comment · Reviewer_RkAM · 2024-08-13
> > **Thanks for the response**
> >
> > First of all, my apology for the late chiming in. My work email changed recently so I didn't receive any notification...
> >
> > Please understand that my overall sentiment is still positive although my score fell on the skeptical side. I really do like the technique and I'm happy to raise my score to 6 for the theoretical merit. Nonetheless, there are still parts of the empirical results that are not convincing to me,
> >
> > Regarding table 3:
> > I'm not fully agreeing with what is considered a success here. I feel like some signal showing the practicality of your method is sorely needed. Maybe the right way to move forward is to set up an implementation of GP without the KeOps backend, so things can be compared on equal footing.
> >
> > Regarding BNN:
> > I agree that there are many strategies for setting up BNN. Perhaps I missed the point on the Laplacian approximation being relevant to your method. However, I disagree that it is irrelevant to compare to other techniques because it would be interesting to show that how your improvement fares against other sota methods.

---

### Author Rebuttal · Authors · 2024-08-05

We thank all reviewers for their reviews and for assessing the paper so positively!
We are grateful that all reviewers praised the clarity of the contribution.
Reviewers FpA9 and X6Aj did not find any particular weaknesses, and we believe that the weaknesses listed by RkAM and 3YyQ might be easy to resolve.

We replied to all reviews in separate threads and look forward to the discussion.

Thank you again, and best wishes,

Authors

---

### Comment · Reviewer_FpA9 · 2024-08-07
**Dear all**

I have read the authors' response and the other reviewers' comments on the submission. I am overall happy with the authors' rebuttal response and would vote for acceptance of this paper.

---

### Decision · Program_Chairs · 2024-09-25

**Decision:**

Accept (spotlight)

**Comment:**

This paper presents a novel method for efficiently computing gradients for functions of large matrices using Lanczos and Arnoldi iterations, tackling challenges in automatic differentiation for tasks such as Gaussian processes, PDEs, and Bayesian neural networks. Implemented in JAX, the approach offers notable improvements in speed and scalability without the need for explicit matrix formation or problem-specific optimizations. Reviewers commend the paper for its strong theoretical contribution, clear presentation, and empirical validation, with only minor criticism regarding the empirical evaluation. I concur with the reviewers' consensus on acceptance.